EMBO
Molecular Medicine

# A skin colonizer disrupts inflammatory and humoral immune defenses in hidradenitis suppurativa

Viviane A Agbogan [1], Florence Bugault[1,11], Laure Guenin-Macé [1,9,11✉], Inta Gribonika [2,10], Cyril Planchais [3], Jean-David Morel [4], Jérémie Delaleu [2], P Juliana Pérez-Chaparro[5], Michael Atlan[6], Maïa Delage[7], Aude Nassif[7], Hugo Mouquet [3], Yasmine Belkaid[2,9], Olivier Join-Lambert [8✉] & Caroline Demangel [1✉]

## Abstract

Hidradenitis suppurativa (HS) is a chronic inflammatory skin disease associated with a polybacterial dysbiosis devoid of a known pathogen. Here, we report that HS patients mount IgA and IgG responses against skin colonizers, notably *Porphyromonas uenonis* (Pu), a rare species selectively enriched in severe disease. In these patients, Pu foci are detected in the epidermis, surrounded by IgA deposits, and anti-Pu IgGs cross-react with self-antigens expressed by healthy keratinocytes. Using healthy human skin explants, we demonstrate that patient-derived Pu can cross an intact epidermal barrier, infect and persist within keratinocytes, triggering their expression of pro-inflammatory mediators. In contrast, topical application of Pu on immunocompetent mice elicit cutaneous and systemic humoral immune responses without tissue infection. These findings uncover an impaired innate immune control of Pu in HS patients, linking keratinocyte infection to skin inflammation and humoral autoimmunity. They underscore the potential of targeting cutaneous dysbiosis as a strategy to limit HS progression.

**Keywords** Skin; Inflammation; Antibodies; Bacteria; Keratinocyte
**Subject Categories** Immunology; Microbiology, Virology & Host Pathogen Interaction; Skin

## Introduction

The skin forms our primary barrier against the outside world and microbial invasion. Commensal bacteria colonizing the skin surface play a pivotal role in this protective function by stimulating immune responses through an intact epidermal barrier (Chen et al, 2018). While keratinocytes (KCs) and tissue-resident T cells are known to mediate microbial control, the role of humoral immunity remains less defined. B cells and plasma cells (PCs) primarily infiltrate the skin in the context of cutaneous inflammatory or autoimmune conditions (Lee and Oh, 2024). However, the presence of immunoglobulin (Ig)A/G/M in human sebum and the stratum corneum suggests that humoral effectors participate in antimicrobial defense (Okada et al, 1988; Metze et al, 1989). In line with this concept, recent murine studies showed that topical exposure to novel commensals induces local PCs producing anti-bacterial IgGs to maintain microbial homeostasis and prevent subsequent infection (Gribonika et al, 2025). These findings raise the possibility that antimicrobial immune defenses may be impaired in chronic inflammatory skin diseases with dysbiosis, such as atopic dermatitis (AD) (Fyhrquist et al, 2019) or hidradenitis suppurativa (HS) (Riverain-Gillet et al, 2020; Guet-Revillet et al, 2017).

HS is a chronic inflammatory disease of pilosebaceous units, typically developing after puberty, with a global prevalence of 0.03–4% and a female predominance (Phan et al, 2020). Disease severity is classically categorized into 3 levels using the Hurley scoring system. The less severe, referred to as HS1, involves recurrent inflamed nodules in the axillae, groin or perineum. Some individuals develop more severe lesions, with draining sinuses and perilesional inflammation (HS2), sometimes involving a whole anatomical region (HS3). The pathogenesis and chronicity of HS remain poorly understood, partly due to the absence of animal models. Unlike AD, the cutaneous dysbiosis in HS is not dominated by a single bacterial species. Instead, microbiome analyses of HS patients' skin revealed variable alterations, with a consistent enrichment of normally rare anaerobes such as *Porphyromonas* and *Prevotella spp.* in lesional and, to a lesser extent, unaffected skin (Riverain-Gillet et al, 2020; Guet-Revillet et al, 2017; Ring et al, 2017, 2019; Świerczewska et al, 2022; Naik et al, 2020).

[1]Institut Pasteur, Université Paris Cité, Inserm U1224, Immunobiology and Therapy Unit, Paris, France. [2]Metaorganism Immunity Section, Laboratory of Host Immunity and Microbiome, National Institute of Allergy and Infectious Diseases, National Institutes of Health, Bethesda, MD 20892, USA. [3]Institut Pasteur, Université Paris Cité, Humoral Immunology Unit, Paris, France. [4]Ecole Polytechnique Fédérale de Lausanne, Institute of Bioengineering, Laboratory of Integrative Systems Physiology, Lausanne, Switzerland. [5]NIAID Microbiome Program, Laboratory of Host Immunity and Microbiome, National Institute of Allergy and Infectious Diseases, National Institutes of Health, Bethesda, MD 20892, USA. [6]Tenon Hospital, Institut Universitaire de Cancérologie, Department of Plastic, Reconstructive and Aesthetic Surgery, Assistance Publique-Hôpitaux de Paris, Paris, France. [7]Institut Pasteur, Université Paris Cité, Medical Center, Paris, France. [8]Univ de Caen Normandie, Univ Rouen Normandie, INSERM, DYNAMICURE UMR 1311, CHU Caen, Department of Microbiology, F-14000 Caen, France. [9]Present address: Institut Pasteur, Université Paris Cité, Metaorganism Unit, Paris, France. [10]Present address: Division of Molecular Hematology, Department of Laboratory Medicine, Lund University, Lund, Sweden. [11]These authors contributed equally: Florence Bugault, Laure Guenin-Mace. ✉E-mail: laure.guenin-mace@pasteur.fr; olivier.join-lambert@unicaen.fr; caroline.demangel@pasteur.fr

**Table 1. Characteristics of serum donors.**

| | HS patients (total) | HS1 | HS2 | HS3 | AD patients | HC |
|---|---|---|---|---|---|---|
| Number | 57 | 29 | 17 | 11 | 7 | 32 |
| Female/Male ratio | 1.11 | 2.57 | 0.30 | 0.56 | 1.33 | 1.28 |
| Age mean (SD) | 32.8 (9.5) | 32.5 (8.5) | 33.1 (9.6) | 33.2 (12.5) | 32.6 (7.6) | 39.2 (11.0) |

Because HS lesions are marked by abundant myeloid and Th1/Th17 cell infiltrates (Jiang et al, 2021), current treatments primarily target pro-inflammatory cytokines (Frew et al, 2019, 2021). Although the TNF-α antibody adalimumab, the first FDA-approved treatment, was a major advance, it benefits only 42–59% of the patients (Lu et al, 2021). Newer antibodies against IL-17A (secukinumab) or IL-17A/F (bimekizumab) have not achieved higher response rates (Kimball et al, 2023; Egeberg and Thyssen, 2023). None of these immunomodulatory treatments fully resolves symptoms in responders, highlighting the need for novel approaches.

While broad-spectrum antibiotics can alleviate symptoms and induce remission in patients with HS (Delage et al, 2023; Join-Lambert et al, 2011, 2016; Chahine et al, 2018), skin dysbiosis remains considered a consequence of the disease rather than a primary triggering factor (Naik et al, 2019). Indeed, the bacteria colonizing the skin of patients with HS are not classical human pathogens. Moreover, the permeability of the epidermal barrier is not significantly altered in patients with HS, even in inflamed areas (Somogyi et al, 2023). Notably, HS skin exhibits increased PC infiltration and a characteristic autoantibody profile in tissue and serum (Hoffman et al, 2018b; Musilova et al, 2020; Gudjonsson et al, 2020; Byrd et al, 2019; Sabat et al, 2023; Hoffman et al, 2018a). Tertiary lymphoid structures with Ig-producing PCs have been found around severe HS lesions (Yu et al, 2024; Lowe et al, 2024), including autoreactive IgG-secreting PCs targeting KCs and promoting autoimmune pathogenesis (Byrd et al, 2019; Carmona-Rivera et al, 2022; Macchiarella et al, 2023; Ross and Ballou, 2022). B-cell depletion with anti-CD20 (rituximab) successfully treated HS in a case report (Takahashi et al, 2018) and showed broad anti-inflammatory effects in HS skin explants (Vossen et al, 2019). These findings suggest a pathogenic role for humoral immune responses. However, their relationship to skin dysbiosis remains unexplored to date.

In this study, we hypothesized that the inflammatory and humoral immune responses observed in HS skin may arise in response to colonizing bacteria. We identify *Porphyromonas uenonis*, uniquely expanding on the skin of patients with severe HS, as a potent inducer of IgA/G responses cross-reacting with KCs. Spatial mapping of live bacteria in patient-derived skin samples and healthy skin explants coated with cultured bacteria revealed that *P. uenonis* actively penetrates the human epidermal barrier. In vitro, *P. uenonis* infects and persists within primary KCs, inducing sustained production of pro-inflammatory cytokines and chemokines. In HS skin, epidermal invasion by *P. uenonis* coincides with IgA production, a response replicated in mice by topical bacterial application. However, it does not result in tissue colonization in this immunocompetent model. These findings suggest that in HS, These findings suggest that in HS, the ability of *P. uenonis* to penetrate the skin, colonize KCs, and elicit inflammatory and humoral responses may play a central role in the development of epidermal immunopathology.

## Results

### Skin-colonizing bacteria induce severity-specific Ig responses in HS patients

To investigate anti-bacterial responses across HS severity levels, we collected pre-treatment serum samples from 29 HS1, 17 HS2, and 11 HS3 patients (Table 1). Age- and sex-matched healthy controls (HC, $n = 32$) and AD patients (AD, $n = 7$) served as comparator groups. Skin microbiome samples from HS patients were also collected for metagenomic profiling and isolation of clinical strains representative of the disease (Table 2). As expected from previous studies (Ring et al, 2019; Riverain-Gillet et al, 2020; Guet-Revillet et al, 2017), HS3 patients displayed a skin microbiota profile marked by increased relative abundances in *Porphyromonas* and *Prevotella spp*, although these remain minor members of the overall microbiota, while in contrast the relative abundance of *Cutibacterium acnes* did not increase (Fig. 1A). Among the anaerobic isolates, a *Porphyromonas spp* was identified as *P. uenonis*, based on an average nucleotide identity (ANI) of 93.8% just below the 95% species-level threshold (Table 2). Eleven other representatives of bacterial strains classically up- or down-modulated in HS skin (Riverain-Gillet et al, 2020; Guet-Revillet et al, 2017; Ring et al, 2017, 2019; Świerczewska et al, 2022; Naik et al, 2020) composed the bacterial isolates collection (Table 2), to which *Lactobacillus reuteri* was added to serve as a control commensal. We then quantified serum IgG, IgA, and IgM binding to live bacteria using a flow cytometry-based assay (Moor et al, 2016).

Principal component analysis (PCA) of serum Ig levels normalized to a reference human serum revealed a heterogeneous profile in HS patients, distinct from the more homogeneous and overlapping profiles of HC and AD patients (Fig. 1B). This heterogeneity was primarily driven by HS3 patients, who showed elevated anti-bacterial IgA/G responses compared to HS1/2 (Fig. 1B, C). Notably, all HS patients exhibited IgG reactivity to *P. uenonis* and *P. bivia*, with antibody levels increasing with disease severity (Fig. 1C, D). IgA responses to *P. bivia* and *Prevotella disiens* showed a similar trend (Figs. 1C, D and EV1A). However, IgAs targeting *P. uenonis* were restricted to HS3 patients. No such responses were detected in AD patients, suggesting HS-specific reactivity rather than a general marker of chronic skin inflammation. Of note, HS1/2 patients displayed IgA/G responses to *Staphylococcus lugdunensis*, absent in HS3 and control groups (Fig. 1C and EV1A). Although the targeted bacterial species were similar across sexes, women exhibited higher overall IgA/G levels (Fig. EV1B). Together, these findings showed that skin-colonizing

**Table 2. Bacteria collection.**

| Name | Growth atmosphere | Relative enrichment[a] | Source[b] |
|---|---|---|---|
| Porphyromonas uenonis[c] | Anaerobic | HS + | Met |
| Staphylococcus lugdunensis | Aerobic | HS + | Met |
| Prevotella bivia | Anaerobic | HS + | Met |
| Prevotella disiens | Anaerobic | HS + | Met |
| Cutibacterium acnes | Anaerobic | HS + | Met |
| Fusobacterium nucleatum | Anaerobic | HS + | Met |
| Parvimonas micra | Anaerobic | HS + | Met |
| Streptococcus anginosus | $CO_2$ / Anaerobic | HS + | Met |
| Actinomyces radingae (Schaalia radingae)[d] | $CO_2$ / Anaerobic | HS + | Met |
| Actinomyces turicensis (Schaalia turicensis)[d] | $CO_2$ / Anaerobic | HS + | Met |
| Dermabacter hominis | Aerobic | HS - | Met |
| Corynebacterium tuberculostearicum | $CO_2$ / Anaerobic | HS - | Met |
| Lactobacillus reuteri (Limosilactobacillus reuteris)[d] | $CO_2$ / Anaerobic | HC | CIP, #101887 T |

[a]Relatively enriched or depleted in HS skin compared to healthy skin (HS+ or HS-, respectively); representative of healthy skin (HC) (Guet-Revillet et al, 2017; Riverain-Gillet et al, 2020).
[b]Clinical isolates harvested and authenticated by OJL in the frame of the MetHS study (Met) or obtained from the Centre de Ressources Biologiques de l'Institut Pasteur (CIP).
[c]Based on MALDI-TOFF analysis and sequencing, showing P. uenonis DSM 23387 as the first hit with an average nucleotide identity (ANI) of 93.8%.
[d]Basionym (new name).

bacteria elicit robust humoral immune responses in HS, with severity-specific Ig profiles and heightened IgA/G responses to *P. uenonis* and *P. bivia* in severe cases.

## Autoreactivity of anti-bacterial antibodies

Recent studies reported the presence of autoantibodies in HS patients, including IgGs targeting KCs (Yu et al, 2024; Carmona-Rivera et al, 2022; Macchiarella et al, 2023; Ross and Ballou, 2022; Oliveira et al, 2023). Consistent with these findings, sera collected from HS3 patients showed significant IgG reactivity to the superficial epidermis of healthy skin sections, whereas IgA autoreactivity was undetectable (Fig. 2A). IgGs also reacted strongly to primary KCs isolated from healthy individuals (Fig. 2B), suggesting a recognition of KC components. In ELISAs assessing IgG binding to common autoantigens, HS3 patient sera exhibited a modest yet statistically significant increase in reactivity to lysozyme (LZ) (Figs. 2C and EV2A). To assess whether LZ binding was mediated by cross-reactive anti-bacterial IgGs, serum pools from HC and HS3 patients were immunoadsorbed with *P. uenonis*, *P. bivia*, or buffer as a control (Fig. 2D). Effective depletion of anti-bacterial IgGs was confirmed by a >90% reduction in IgG binding to bacteria, as measured by flow cytometry (Fig. EV2B). We then compared reactivity to LZ in depleted versus non-depleted sera by ELISA. Figure EV2C shows the ELISA titration curves obtained for

HS3 and HC serum pools following depletion with *P. uenonis* or *P. bivia*, while the graphs in Fig. 2D present the mean AUC in each condition and the corresponding reactivity loss for each pool. Depletion of anti-*P. uenonis* antibodies in HS3 sera, but not HC sera, resulted in a significant decrease (58 ± 13%) in IgG reactivity to LZ. In contrast, depletion of anti-*P. bivia* antibodies had no significant effect in either HS3 or HC sera (Fig. 2D). Collectively, these results indicate that in HS3 patients, *P. uenonis*–reactive IgGs directly contribute to autoreactivity against epidermal antigens.

## *Porphyromonas* invades the epidermis in severe HS

To investigate the functional relevance of anti-bacterial IgA/G responses, we analyzed Ig subclass distribution. IgGs against *P. uenonis* or *P. bivia* were predominantly IgG1 (Fig. 3A), consistent with classical IgG responses involving complement activation and opsonization. With respect to IgAs, the IgA1 subclass normally predominates in serum while IgA2 and secretory IgA (sIgA) are the prominent IgAs in human secretions. Compared to HC, HS3 patient sera exhibited abnormally elevated levels of of anti-bacterial IgA2 and significantly higher levels of sIgA, as shown by a secretory component capture assay (Fig. 3B). Depletion of *P. uenonis* from HS3 sera showed a non-significant trend toward reduced sIgA, which was not observed for *P. bivia* (Fig. 3C). Although not statistically significant, this trend, together with the robust anti-bacterial IgA responses shown in Fig. 1C and their altered IgA1/IgA2 profile, raised the possibility that *P. uenonis* may have crossed the skin barrier in HS3 patients.

To test this hypothesis, we collected skin samples from three additional HS3 patients and HC (Table 3). Histological examination revealed typical HS pathology, including epidermal hyperplasia and epithelial sinuses extending in the dermis and without overt rupture of the epidermal barrier (Fig. EV3A). Using mRNA probes specific for *P. uenonis* and *P. bivia* that were validated for reactivity and specificity on bacterial cell cultures (Fig. EV3B), we assessed the presence of metabolically active bacteria in skin samples by in situ hybridization. Strikingly, *P. uenonis* was consistently found within the epidermis of all 3 HS3 patients but was absent in HC (Fig. 3D). The tissue distribution of *P. uenonis* varied across patients, being primarily localized in the outermost layers in Patient 2, suprabasal layers in Patient 1, and both in Patient 3. In contrast, *P. bivia* was only detected at low levels in one HS3 sample and not in HC (Fig. EV3C). In HS3 patient samples, epidermal invasion by *P. uenonis* correlated with local IgA/G production (Fig. 4). Together, these results identified *P. uenonis* as a skin invader in severe HS, supporting its induction of local and systemic humoral immune responses.

## *Porphyromonas* infects keratinocytes and triggers inflammation

To assess the underlying mechanism and immune consequences of *P. uenonis* invasion, we examined its ability to infect KCs in vitro. Primary KCs from healthy donors were incubated with fluorescently labeled *P. uenonis* or *P. bivia* under anaerobic conditions. Notably, >60% of live KCs were infected by *P. uenonis* after 48 h of bacterial contact, while infection with *P. bivia* was hardly detectable (Fig. 5A). Moreover, contrary to *P. bivia*, KC infection with *P. uenonis* had little effect on their viability (Fig. 5A). Confocal

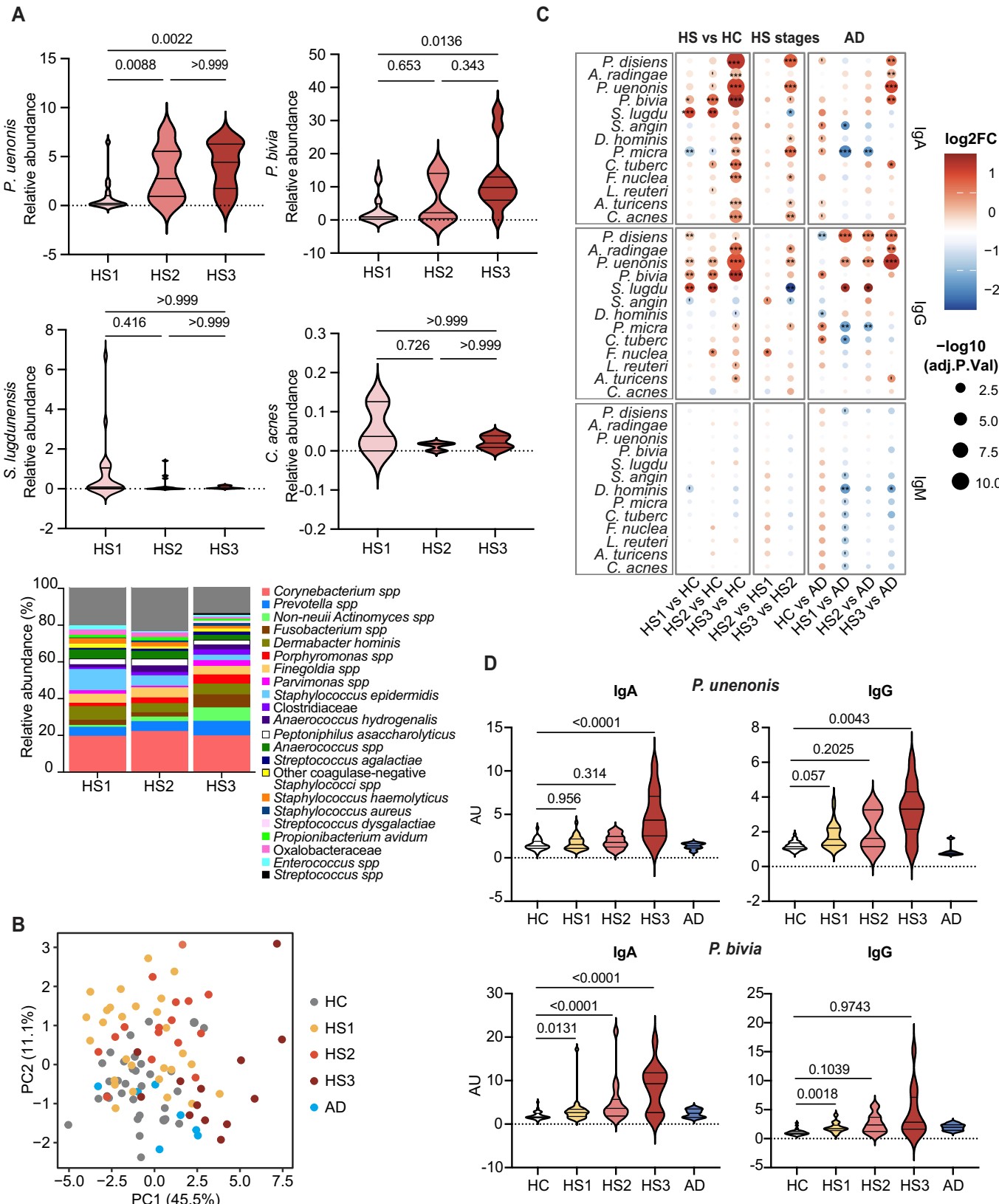

◄ **Figure 1. Skin colonizing bacteria induce severity-specific Ig responses in HS patients.**

(A) Mean relative abundance of *P. uenonis*, *P. bivia*, *C. acnes*, and *S. lugdunensis* in the skin microbiota of HS patients (HS1, $n = 29$; HS2, $n = 17$; HS3, $n = 11$) are shown in relation with disease severity and compared by Kruskal–Wallis test with Dunn's multiple comparison correction. Their relative presence across HS stages is illustrated by stacked bar plots, with RA calculated from metagenomic read counts, re-normalized to 100%, and taxa representing <1% grouped as "Others." (B) PCA of anti-bacterial Igs in sera of HS patients with different severity grades (HS1/2/3), healthy controls (HC) and AD patients. (C) Heatmap comparing the mean levels of anti-bacterial IgAs, IgGs and IgMs in sera of HS1/2/3 patients to those of HC (left), to each other (middle) or those of AD patients (right). *$p < 0.05$, **$p < 0.01$, ***$p < 0.001$, and ****$p < 0.0001$. (D) Relative levels of serum IgA/Gs targeting *P. uenonis* and *P. bivia* in HS1 ($n = 29$), HS2 ($n = 17$), HS3 ($n = 11$) or AD patients ($n = 7$), compared to HC ($n = 32$). Statistical analyses in (B–D) used a moderated, FDR-corrected two-tailed $t$-test (limma). Source data are available online for this figure.

microscopy using Na/K-ATPase surface staining and 3D reconstruction confirmed the intracellular localization of *P. uenonis* (Fig. 5B). Importantly, infected KCs produced elevated levels of pro-inflammatory cytokines, including IL-1β, TNF-α, IL-6 and IL-8, whereas *P. bivia* triggered a comparatively weaker response (Fig. 5C).

To determine whether *P. uenonis* can penetrate an intact epidermal barrier, we used explants of healthy human skin. Tissues were covered with a confluent agar culture of *P. uenonis*, or *P. bivia* as a comparative, then incubated under anaerobiosis for 4 h before being returned to aerobic incubation for 3 days, as described (Maboni et al, 2017) (Fig. 5D). In situ hybridization of bacterial mRNA revealed *P. uenonis* in deep epidermal layers and the underlying dermis (Fig. 5E). This was confirmed using a pan-bacterial control probe (16S mRNA), whereas no signal was detected in control explants or with *P. bivia* (Figs. 5E and EV4). Induction of inflammatory responses was assessed through quantitative analysis of transcripts in tissue lysates (Fig. 5F). IL-1β expression was not increased in explants exposed to bacteria, most likely because the peak of cytokine expression had already passed after 3 days of stimulation. In contrast, transcripts of IL-8 and IL-6, as well as of the chemokine CCL20 known to be induced by cytokines such as IL-1β (Cai et al, 2019), were strongly upregulated, indicating KC activation and the induction of inflammatory responses. These findings revealed the invasive and pro-inflammatory capacity of *P. uenonis* in healthy human skin.

### Topical association with *P. uenonis* triggers humoral immune responses in vivo

To assess the impact of *P. uenonis* on cutaneous and systemic responses in physiological conditions, we established a topical association (TA) model in immunocompetent wild-type mice. We examined bacterial ability to infect the skin and stimulate local and systemic antibody production following TA. Mice received TA of *P. uenonis* or *P. bivia* on the ear skin for 4 days using a procedure previously shown to support low-level, natural colonization by skin-resident bacteria without compromising the barrier (Gribonika et al, 2025; Naik et al, 2015; Enamorado et al, 2023) (Fig. 6A). The bacteria were undetectable in skin swabs or tissue digests targeting colonized appendages at 7, 14, and 60 days post-TA. Consistent with their anaerobic biology, this suggests that *P. uenonis* or *P. bivia* either do not survive on intact skin surface or persist below the detection limit of culture-based methods. We then assessed anti-bacterial humoral immune responses at 14- and 60-days post-TA. Consistent with findings in HS3 patients, we observed significantly elevated levels of *P. uenonis*-specific serum IgG1, IgG2b, and IgA targeting at day 60 (Fig. 6B), while in contrast, mouse skin association with *P. bivia* triggered only an

IgG2b response (Fig. EV5A). In addition, mice associated with *P. uenonis* exhibited increased total serum SIgA levels at 60 days post-TA (Fig. 6C).

To determine whether *P. uenonis* had penetrated the ear epidermis following TA, we analyzed skin sections at days 3 and 60 post-TA using in situ hybridization with the *P. uenonis*-specific probe, or a scrambled 16S probe as a control. No bacterial signal was detected at either time point (as shown in Fig. EV5B at Day 3 post-TA), indicating a lack of sustained tissue invasion. Nevertheless, primary mouse KCs were as susceptible as human KCs to *P. uenonis* ex vivo (Fig. EV5C), and anti-*P. uenonis* IgA/G responses developed in vivo (Fig. 6B), suggesting that skin infection occurred at sites or times not captured, likely due to experimental limitations.

In line with this hypothesis, CD138⁺ PCs and CD19⁺ B cells were significantly increased in the skin of *P. uenonis*-associated mice at both day 14 (Fig. 6D) and day 60 (Fig. 6F) compared to controls. Three-dimensional reconstruction of perpendicular skin sections localized recruited PCs in the epidermal compartment, as defined by CD49f⁺ KC staining (Fig. 6D). ELISPOT analysis confirmed the antigen specificity of these PCs, revealing significant reactivity towards *P. uenonis* antigens in bacteria-exposed mice, but not in control mice (Fig. 6E). PC recruitment was specific to TA with *P. uenonis*, as it was absent in control mice (Fig. 6D, F, G) and was not induced by TA with *P. bivia* (Fig. EV5D). Flow cytometry analysis of ear-derived cells further corroborated these findings, showing increased frequencies of IgA⁺ PCs in *P. uenonis*-associated mice 60-days post-association (Fig. 6H). Exposure to new commensals has been reported to elicit the formation of germinal center (GC)-like structures within the skin, supporting local humoral responses (Gribonika et al, 2025). However, neither GCB cells nor memory B cells were detected in the skin of mice 60 days after TA with *P. uenonis* (Fig. EV5E, F), suggesting that the epidermal PCs originated from peripheral lymphoid tissues rather than being generated locally. Collectively, these findings indicated that in immunocompetent hosts, *P. uenonis* activates systemic B cell responses leading to the recruitment of antigen-specific PCs to the epidermis, without compromising the epidermal barrier or infecting the tissue. Anti-bacterial Ig responses were overall comparable in TA mice and HS3 patients, yet bacteria failed to colonize the skin of TA mice, suggesting that HS3 patient susceptibility to *P. uenonis* is driven by factors beyond humoral immunity.

## Discussion

Whether and how cutaneous dysbioses contribute to chronic inflammatory skin diseases remains poorly defined (Chen et al, 2018).

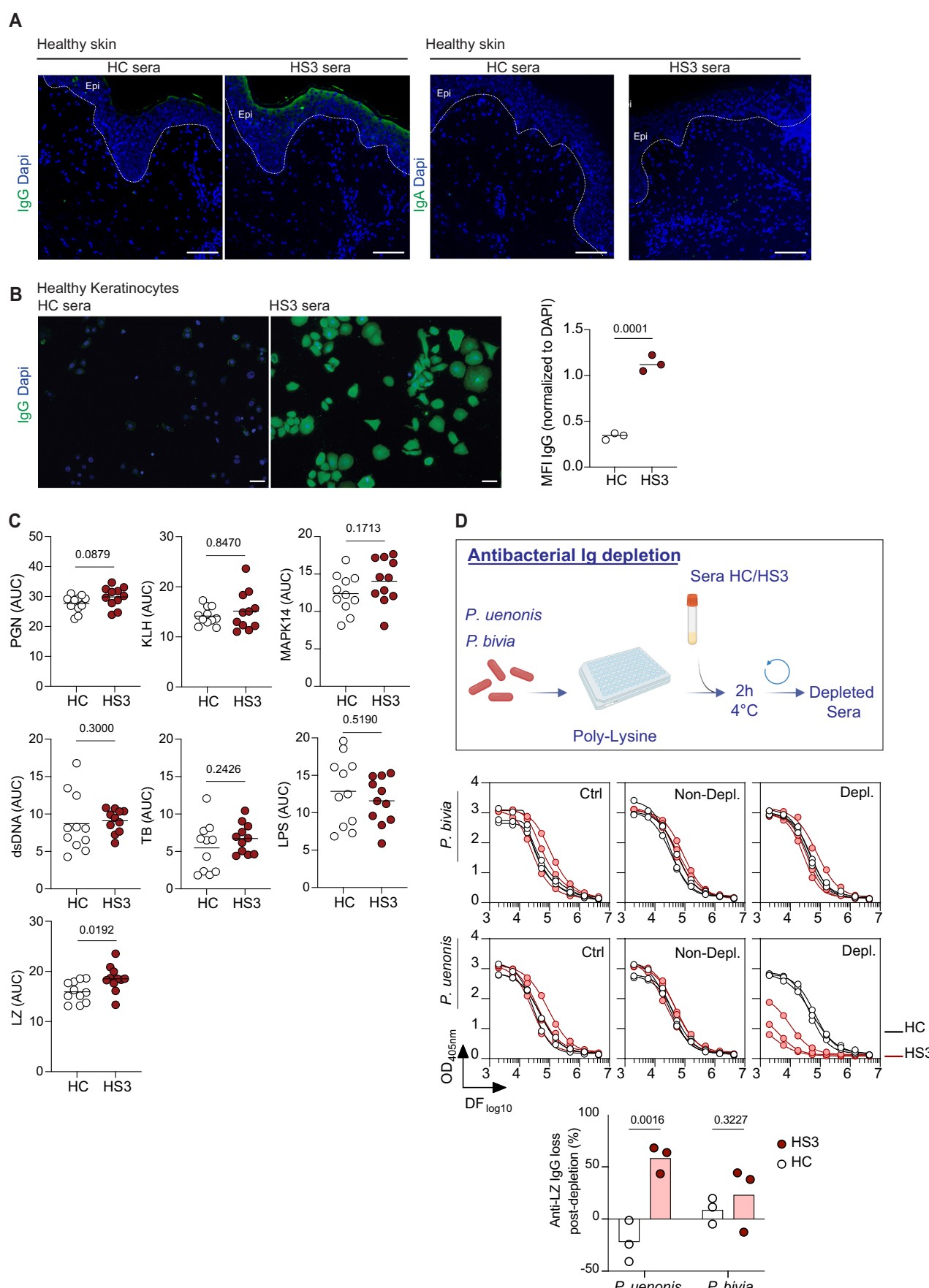

◀ **Figure 2. Anti-bacterial IgGs cross-react with skin antigens.**

(A) IgG (left) and IgA (right) reactivities of HC and HS3 serum pools towards healthy skin sections are shown. Scale bar: 100 µm. The white dotted line indicates the separation between epidermis (Epi) and dermis (Dermis). (B) Same as in (A) towards healthy donor-derived KCs, with quantification of IgG signals normalized to Dapi. Scale bar: 50 µm. Mann–Whitney Statistical test, n = 3. (C) Dot plots illustrating the reactivity of HC and HS3 sera (n = 11) against peptidoglycan (PGN), Keyhole limpet hemocyanin (KLH), MAPK14, lipopolysaccharide (LPS), double-stranded DNA (dsDNA), tubulin (TB), and lysozyme (LZ). Data were area under the curve (AUC) values determined by ELISA with serially diluted sera. Mann–Whitney statistical test. (D) Methodology for depleting HC and HS3 serum pools on bacteria (top). Panels below show the differential reactivity of HC and HS3 sera pools against LZ after depletion on *P. uenonis* or *P. bivia*, expressed as AUC values from ELISA with serially diluted sera, and as the mean percentage of IgG loss in bacteria-depleted pools (n = 3) compared to non-depleted sera, analyzed by two-way ANOVA with Sidak's multiple comparisons test. Source data are available online for this figure.

In the context of HS, establishing a causal link between the skin microbiota and pathology has been hindered by the polymorphic and variable composition of the colonizing flora, and by the limited knowledge regarding emerging taxa such as *Porphyromonas* spp. The *Porphyromonas* genus comprises several anaerobic Gram-negative species found across a wide range of hosts and environments, yet remains understudied. Most human-associated *Porphyromonas spp* are present in both healthy and diseased states (Guilloux et al, 2021). *P. gingivalis* represents a notable exception, with a markedly increasing abundance in the oral cavity under pathological conditions (Lamont and Jenkinson, 1998; Sheridan et al, 2024). *P. gingivalis* has been directly implicated in the pathogenesis of periodontitis and was recently proposed to promote orodigestive carcinogenesis (Acuña-Amador and Barloy-Hubler, 2020). Of particular relevance to the present work, *P. gingivalis* has been shown to localize within buccal epithelial cells, where it is capable of intracellular persistence and intercellular dissemination (Yilmaz et al, 2006; Hendrickson et al, 2009). Here, we demonstrate that *P. uenonis*, a species phylogenetically distant from *P. gingivalis*, possesses the capacity to penetrate intact skin explants and infect deep-seated KCs. Outside the context of HS, *P. uenonis* has been reported only sporadically and at low abundance in the human microbiota, without clear tropism for a specific organ or anatomical site, and it has only exceptionally been associated with active infections (Acuña-Amador and Barloy-Hubler, 2020). To better understand its potential pathogenicity in HS, future studies should assess the genetic and transcriptomic diversity of *P. uenonis* isolates from HS patients. It will be critical to determine whether these isolates differ from those collected from healthy individuals or represent novel strains. Comparative analyses with other *Porphyromonas* spp, particularly *P. gingivalis*, whose virulence depends on tissue invasion factors like gingipains and pro-inflammatory components, may help identify determinants of pathogenicity in the skin environment.

Multiple lines of evidence indicate that KCs play a leading role in the initiation and perpetuation of skin inflammation in HS. KCs from perilesional skin produce elevated levels of the pro-inflammatory cytokines such as TNF-α, IL-1β, and the chemokine CCL2, mirroring the inflammatory profile of KCs in lesional skin (Schell et al, 2023; Dajnoki et al, 2022; Lima et al, 2016). Moreover, ex vivo studies have shown that KCs isolated from HS patients exhibit an intrinsically pro-inflammatory phenotype and an altered profile of antimicrobial peptide expression compared to KCs from healthy donors (Hotz et al, 2016). These data suggest that KCs from HS patients may be more permissive to bacterial invasion and more prone to amplify inflammatory responses upon infection. This pro-inflammatory predisposition may facilitate skin penetration by *P.*

*uenonis*, enabling the establishment of infectious foci that activate humoral responses. Interestingly, although mean abundances of *P. uenonis* in the skin microbiota were similar in HS2 and HS3 patients, only HS3 patients developed anti-*P. uenonis* IgA responses (Fig. 1A–C). This discrepancy may reflect the emergence of more virulent strains of *P. uenonis* in HS3, and/or specific defects in barrier defense or bacterial clearance in these individuals. The failure of clinical *P. uenonis* to invade the epidermis of immunocompetent mice supports the possibility of immune deficiencies in HS3 patients. However, further studies will be necessary to dissect the relative contributions of bacterial virulence factors and host-specific defects in determining susceptibility to epidermal invasion.

In our experiments using *P. uenonis*-associated mice, we did not detect local germinal center B cells, suggesting that PCs infiltrating the epidermis originated from peripheral lymphoid organs rather than developing locally. Notably, trafficking of these PCs to the ear epidermis occurred in the absence of tissue invasion, highlighting the bacterium's ability to mobilize humoral immune responses through topical exposure. We ruled out the possibility that murine KCs are intrinsically resistant to *P. uenonis* infection (Fig. EV5C). Findings in mice topically exposed to *P. uenonis* therefore uncouple KC infection from the local recruitment of PC to the skin. Instead, they suggest that in the context of a competent immune system, *P. uenonis* is contained at the surface but still elicits local and systemic IgA responses. The presence of anti-*P. uenonis* IgA-producing PCs in the epidermal layers points to a potential role of these cells in mediating protective immunity.

Multiple studies have identified autoantibody responses in sera from patients with HS (Oliveira et al, 2023; Macchiarella et al, 2023; Carmona-Rivera et al, 2022; Byrd et al, 2019; Mulani et al, 2018; Yu et al, 2024). Of particular relevance to our work, Yu et al. recently described tertiary lymphoid structures in the dermis of HS lesions, enriched in PCs producing antibodies against KCs (Yu et al, 2024). Consistent with these findings, we observed robust IgG reactivity from HS3 patient sera to KCs, both in vitro and ex vivo and identified LZ, a bacteriolytic enzyme constitutively expressed by KCs (Papini et al, 1981), as a cross-reacting autoantigen. However, the contribution of anti-bacterial IgGs to LZ binding was partial (Fig. 2D), suggesting that mechanisms other than cross-reactivity may underlie the link between *P. uenonis* infection and auto-reactivity. A particularly promising hypothesis to test is the reactivity with citrullinated proteins. Indeed, autoantibodies against citrullinated antigens have been detected in HS patient sera (Byrd et al, 2019), and both the expression and enzymatic activity of Peptidylarginine Deiminases (PADs, the enzymes mediating citrullination) were elevated in HS skin compared to HC.

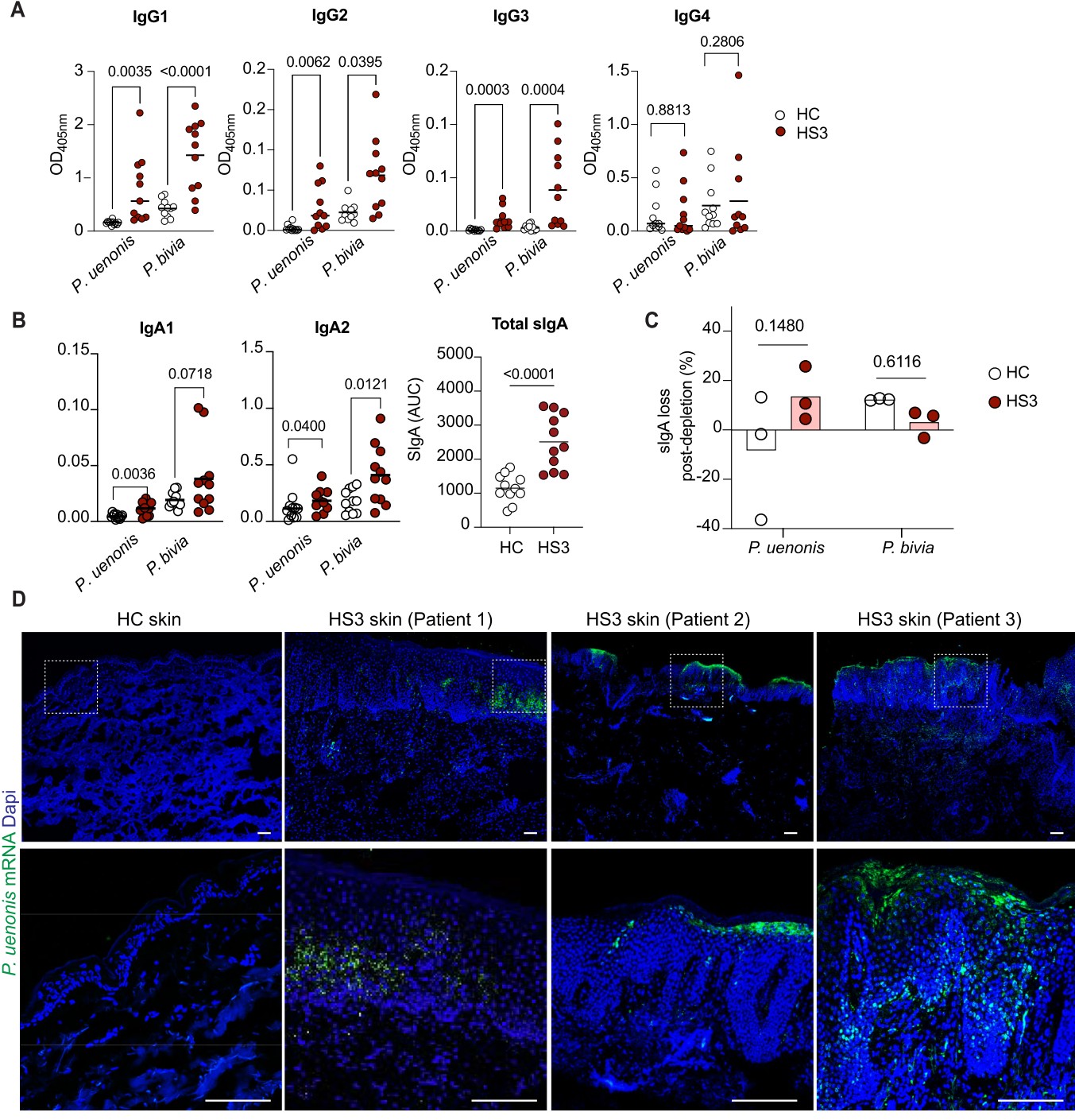

**Figure 3. *Porphyromonas uenonis* invades the epidermis in severe HS.**

(A, B) Subtype analysis of serum IgGs (A) and IgAs (B) targeting *P. uenonis* or *P. bivia* in HS3 patients and healthy controls (HC). Total sIgA levels in HC and HS3 sera, expressed as area under the curve (AUC), are also shown. Statistical analyses used a two-tailed Mann–Whitney test, $n = 11$. (C) Effect of anti-bacterial depletion on sIgA levels. Data were mean % of sIgA loss following depletion on *P. uenonis* or *P. bivia* in three pools of HC and HS3. Statistical analysis used a two-way ANOVA and the Sidak multiple comparisons test. (D) Representative images of skin samples from six HC and three HS3 patients hybridized with an anti-*P. uenonis* mRNA probe, with nuclei counterstained with DAPI. Dotted square indicates the magnified area (bottom panel, Scale bars: 500 μm). Source data are available online for this figure.

Intriguingly, *Porphyromonas gingivalis* is the only known prokaryote that encodes a PAD enzyme. Bacterial PAD was shown to citrullinate both bacterial and human α-enolase, which is abundantly expressed by KCs (Tohgasaki et al, 2018). Citrullination

has recently been implicated in stimulating T cells and breaking down immune tolerance in rheumatoid arthritis (Curran et al, 2023). Thus, determining whether clinical isolates of *P. uenonis* possess PAD activity could clarify whether bacteria-driven

**Table 3. Characteristics of skin donors.**

|  | HS3 patients | Healthy controls |
|---|---|---|
| Number | 3 (Males) | 6 (4 Females + 2 males) |
| Age mean (SD) | 38.6 (5.6) | 34.7 (5.2) |

citrullination of KC proteins may also contribute, directly or indirectly, to the development of autoreactivity in HS. Importantly, anti-bacterial PCs may also contribute to immunopathology via antibody-independent mechanisms (Dang et al, 2014). For instance, in murine *Trypanosoma cruzi* infection, CD19$^+$CD138$^+$ PCs were identified as a major source of pro-inflammatory IL-17 (Bermejo et al, 2013). Similarly, in the gut, PCs can produce iNOS and TNF-α in a microbiota-dependent manner, which in turn promotes IgA production (Fritz et al, 2011). By analogy, humoral immune responses driven by *P. uenonis* may perpetuate inflammation initiated by infected KCs through direct secretion of inflammatory mediators (Gudjonsson et al, 2020), establishing a feed-forward loop that amplifies tissue damage.

In conclusion, we demonstrate that *P. uenonis* is capable of crossing the intact epidermis in patients with severe HS and propose that the chronic activation of the skin's immune defenses, driven by microbial invasion, contributes to immunopathology. While this study focused on *P. uenonis*, our anti-bacterial Ig analysis also suggests a possible association between *Staphylococcus lugdunensis* and milder HS phenotypes, warranting further investigation. Collectively, these findings support a model in which primary defects in immune surveillance at the skin-microbiota interface allow for pathogenic colonization and chronic immune activation. This has important therapeutic implications: rather than broadly suppressing inflammation, strategies should consider restoring microbial control. Our study argues for targeted antimicrobial approaches as adjuncts to immunomodulation, with the potential to synergize with anti-inflammatory biologics and reduce the risk of microbial-driven flares.

# Methods

### Reagents and tools table

| Reagent/resource | Reference or source | Identifier or catalog number |
|---|---|---|
| **Experimental models** | | |
| HS patient sera | METHS study (Institut Pasteur, approval code N° IDRCB: 2011-A00536-35, 2011-14) | Table 1 |
| HC sera | ICAReB platform (Institut Pasteur) | Table 1 |
| AD patient sera | Tebu-bio | Cat#279CUSTOM-T9-1092 |
| Bacterial isolates | (Guet-Revillet et al, 2017; Riverain-Gillet et al, 2020) | Table 2 |
| HS patient and HC skin | CoSimmGEN study (Institut Pasteur, N°2010-006) | Table 3 |
| Primary human KCs | ATCC | Cat# ATCC-PCS-200-011 |

| Reagent/resource | Reference or source | Identifier or catalog number |
|---|---|---|
| Human skin samples | Genoskin Inc | NativeSkin® |
| Mice | NIAID-Taconic exchange program (mouse strain name Tac 8478) | CD45.1 (B6.SJL-Ptprca Pepcb/BoyJ). |
| **Culture media** | | |
| GenBag anaer (bacteria) | Biomérieux | Cat# 45534 |
| Agar medium (bacteria) | Biomérieux | Cat# 418229 |
| EpiLife (Human KCs) | Invitrogen | Cat# EPI-500-CA |
| EpiLife Growth Supplement | Invitrogen | Cat# S-001-5 |
| KC growth medium | PromoCell | Cat# C-20011 |
| Penicillin/streptomycin (P/S) | Invitrogen | Cat# 15140-122 |
| Amphotericin B | Sigma | Cat# A2942-20ml |
| **IF microscopy reagents** | | |
| Superfrost Plus slides | Thermo Fisher Scientific | Cat# 22-037-246 |
| ProLong Diamond mounting solution | Thermo Fisher Scientific | Cat# P36965 |
| Backlight Red bacterial stain | Thermo Fisher Scientific | Cat# B35001 |
| **Molecular biology reagents** | | |
| DNA Easy UltraClean microbial kit | Qiagen | Cat# 10196-4 |
| Nextera XT DNA Library Prep Kit | Illumina | Cat# FC-131-1096 |
| **Antibodies and reagents for IF/ELISA/ELISPOT/FC** | | |
| Mouse IgA ELISA kit | Assay Genie | Cat# MOES01461 |
| Goat Anti-Human IgG, Fcγ Secondary-HRP | Jackson ImmunoResearch | Cat# 109-035-098 |
| Mouse anti-Human IgG1 Fc Secondary-HRP | Thermo Fisher | Cat# A-10648 |
| Mouse anti-Human IgG2 Fd Secondary-HRP | Thermo Fisher | Cat# 05-0520 |
| Mouse anti-Human IgG3 Hinge Secondary-HRP | Thermo Fisher | Cat# 05-3620 |
| Mouse anti-Human IgG4 Fc Secondary-HRP | Thermo Fisher | Cat# A-10654 |
| Goat Anti-Human Serum IgA, α Chain Secondary-HRP | Jackson Immunoresearch | Cat# 109-035-011 |
| Mouse Anti-Human IgA1-HRP | SouthernBiotech | Cat# 9130-05 |
| Mouse Anti-Human IgA2-HRP | SouthernBiotech | Cat# 9140-05 |
| ABTS solution | AAT Bioquest | Cat# 11001 |
| Goat anti-CD138/Syndecan-1 | Fisher Scientific | Cat# PA5-47395 |

| Reagent/resource | Reference or source | Identifier or catalog number |
|---|---|---|
| Mouse anti-human IgA | Proteintech | Cat# 60099-1-IG |
| Rabbit anti-human IgG | Fisher Scientific | Cat# MA5-42729 |
| Donkey Anti-Goat Alexa Fluor 55 | Invitrogen | Cat# A-21432 |
| donkey anti- rabbit Alexa Fluor 647 | Invitrogen | Cat# A-31573 |
| Goat anti-mouse Alexa Fluor 488 | Invitrogen | Cat# A-11001 |
| Anti-Na$^+$/K$^+$ ATPase | Abcam | Cat# AB76020 |
| Invitrogen LIVE/DEAD Fixable Blue Dead Cell Stain Kit | Fisher Scientific | Cat# 17428262 |
| Zombie Nir viability dye | BioLegend | Cat #423105 |
| Human serum depleted of IgA/IgM/IgG | Sigma-Aldrich | Cat# S5393-1VL-PW |
| Anti-GFP recombinant human antibodies | Bio-Rad | Cat# HCA189, HCA191, HCA192 |
| Human serum of reference | Sigma-Aldrich | ERM®- DA470k/IFCC |
| Rat anti-mouse CD138-PE | BD Pharmingen | Cat# 561070 |
| Rat anti-mouse CD73-Pe-Cy7 | BioLegend | Cat# 127223 |
| Rat anti-mouse CD273 | BD Pharmingen | Cat# 560086 |
| Rat anti-mouse CD45-BUV396 | BD Horizon | Cat# 564279 |
| Rat anti-mouse IgD-PerCP-Cy5.5 | BioLegend | Cat# 405709 |
| Rat anti-mouse/human GL-7-eF450 | eBioscience | Cat# 48-5902-82 |
| Rat anti-mouse B220-PECF594 | BD Horizon | Cat# 562313 |
| Rat anti-mouse CD19-BV785 | BioLegend | Cat# 115543 |
| Rat anti-mouse CD19-APC | BD Pharmingen | Cat# 561738 |
| Rat anti-mouse CD49f-eF450 | eBioscience | Cat# 48-0495-82 |
| Rat anti-mouse IgG2b-FITC | BD Pharmingen | Cat# 56-0042-82 |
| Rat anti-mouse CD138-PE | BD Pharmingen | Cat# 561070 |
| Rat anti-mouse IgA-FITC | BD Pharmingen | Cat# 559354 |
| 96-well MaxiSorp microtiter plates Nunc | Fisher Scientific | Cat# 11530627 |
| Goat anti-mouse IgG1-AP | Southern Biotechnology, Birmingham, AL | Cat# 1071-04 |
| Goat anti-mouse IgA-AP | Southern Biotechnology, Birmingham, AL | Cat# 1040-04 |

| Reagent/resource | Reference or source | Identifier or catalog number |
|---|---|---|
| Goat anti-mouse IgG2b-AP | Southern Biotechnology, Birmingham, AL | Cat# 1091-04 |
| Goat anti-mouse IgG2c-AP | Southern Biotechnology, Birmingham, AL | Cat# 1079-04 |
| Goat anti-mouse Ig-AP | Southern Biotechnology, Birmingham, AL | Cat#: 1010-04 |
| Elispot 96-well plates | Millipore | MultiScreen HTS |
| Sigmafast BCIP/NBT B5655 | Sigma-Aldrich | Cat# B5655 |
| **ISH** | | |
| RNAscope kit | Bio-Techne, Minneapolis | Cat# 323100 |
| *P. uenonis* probe | Bio-Techne, Minneapolis | Cat# 838331 |
| *P. bivia* probe | Bio-Techne, Minneapolis | Cat# 1241421-C1 |
| UBC probe | Bio-Techne, Minneapolis | Cat# 320861 |
| *E. coli DapB* probe | Bio-Techne, Minneapolis | Cat# 320871 |
| **QRT- PCR reagents** | | |
| Precellys® tissue homogenizer + beads | Bertin Technologies | Cat# 000911-LYSK0-A.0 |
| RNeasy® Fibrous Tissue Mini Kit | Qiagen | Cat# 74704 |
| High-Capacity cDNA Reverse Transcription Kit | Applied Biosystems | Cat# 4368814 |
| Power SYBR Green PCR Master Mix | Applied Biosystems | Cat# 4367659 |
| PCR primers | This study | Table 4 |
| **Chemicals, enzymes and other reagents** | | |
| Peptidoglycan from *B. subtilis* | Invivogen | Cat# tlrl-pgnb3 |
| Dispase | STEMCELL Technologies Inc | Cat# 07913 |

## Methods and protocols

### Human samples

Skin microbiome and serum samples of HS patients were collected in the frame of the METHS study (Institut Pasteur, approval code N° IDRCB: 2011-A00536-35). Skin samples were collected via the CoSimmGEN study (Institut Pasteur, N°2010-006). Both studies were approved by the French Ethics Committee (Comité de Protection des Personnes, Paris, France), the competent health authority (Agence Nationale de Sécurité des Médicaments et des produits de santé, Saint-Denis, France), and the French Data Protection Agency (Commission Nationale de l'Informatique et des Libertés, Paris, France). The experiments conformed to the principles set out in the WMA Declaration of Helsinki and the Department of Health and Human Services Belmont Report. The number and characteristics of all donors are provided in Tables 1 and 3. Cohort size was determined by the number of HS patients diagnosed during the study period (2011–2014). Inclusions were established prospectively, and no samples nor data were excluded from the study. Participants were anonymized using identification numbers and provided written informed consent before inclusion. Inclusion criteria were an active

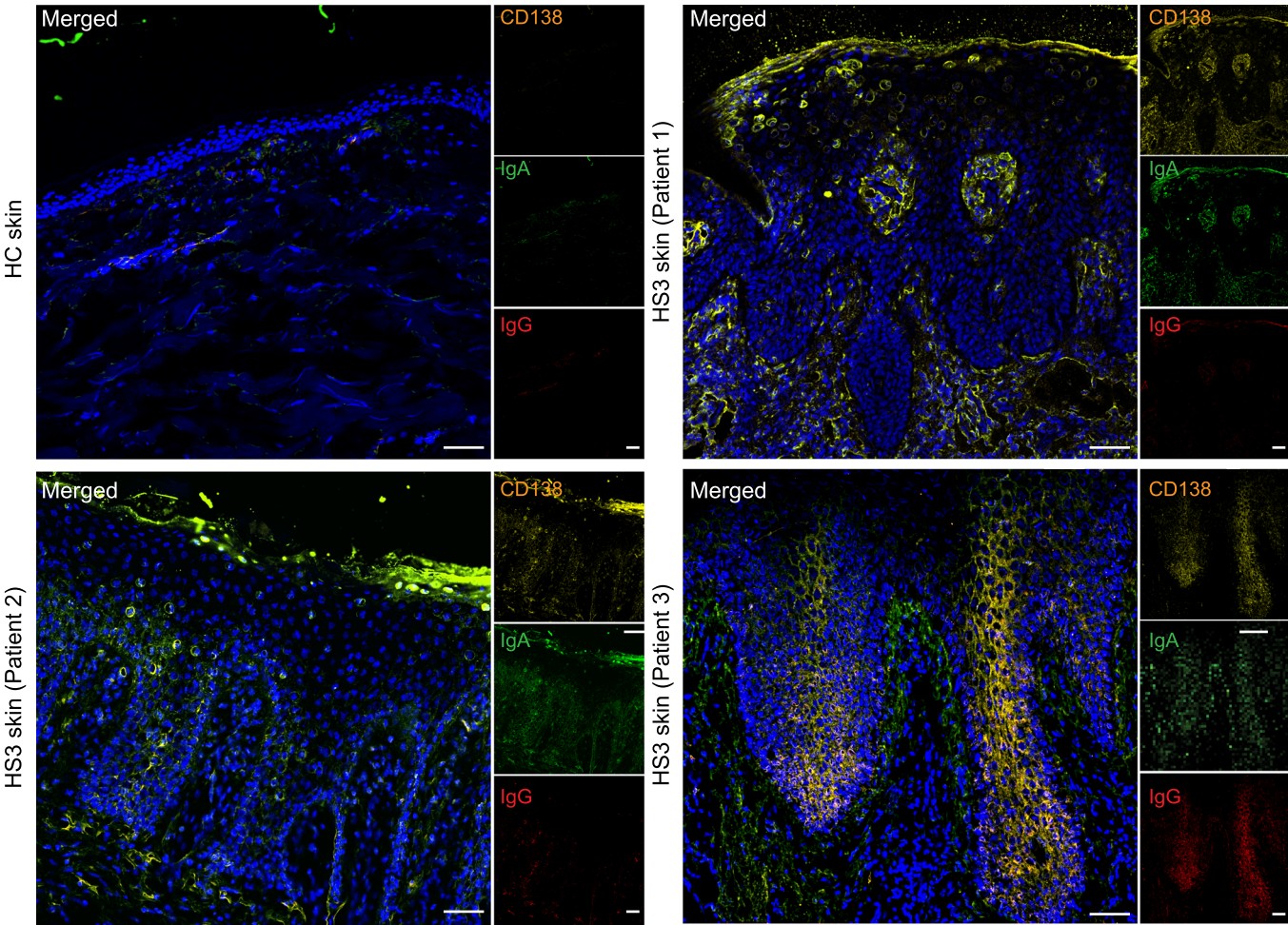

**Figure 4. Epidermal invasion coincides with the local production of IgA/Gs.**

Representative confocal images of skin sections from HC and three HS3 patients stained with Dapi, CD138, IgA, and IgG. Scale bars: 200 μm. Source data are available online for this figure.

HS diagnosis and age of 18 years or older. Exclusion criteria were antibiotic or immunosuppressive treatment (including corticosteroids) during the 2 months before inclusion, pregnancy, chronic liver or kidney disease, cancer or hematological disease, immune deficiency, personal or familial history of chronic inflammatory rheumatologic or intestinal disorder and xylocaine allergy. The serum library was managed by the Centre for Translational Science, Clinical Investigation and Access to Bioresources (ICAReB) platform (Institut Pasteur), which provided serum samples from age- and gender-matched HC. Serum samples from AD patients were purchased from Tebu-bio (Cat#279CUSTOM-T9-1092). Serum samples (1 mL) were stored at −80 °C until use. Skin samples from HC and HS3 patients were collected from surgical resections during plastic surgery or surgical removal of HS lesions, respectively, at Tenon Hospital (Institut Universitaire de Cancérologie, Department of Plastic, Reconstructive and Aesthetic Surgery, Assistance Publique-Hôpitaux de Paris, France). Inclusion criteria required participants to be >18 years old, provide written consent and have an active HS diagnosis or undergo reconstructive surgery for HC donors. Samples were fixed with 4%

paraformaldehyde in PBS overnight, cryopreserved in 30% sucrose for 24 h, embedded in optimal cutting temperature (OCT) compound and stored at −20 °C. Cryostat sections (8 μm) were placed on Superfrost Plus slides (Thermo Fisher Scientific) and stored at −20 °C until analysis.

### Bacterial collection and identification

Microbiota samples were collected from the lesions of the 57 HS1/3 patients and their metagenomic profiles were compared with those of unaffected skinfolds, in an enlarged cohort of 65 patients (Guet-Revillet et al, 2017). Draining lesions were sampled by swabbing pus, and non-draining lesions by punch biopsy or needle aspiration under sterile conditions. Swabs were used for both metagenomic analysis and seeding of cultures. Colonies isolated on agar plates were identified by matrix-assisted laser desorption/ionization time-of-flight mass spectrometry (MALDI-TOF) and genome sequencing. Isolates representing HS-modulated species were frozen for preservation. When needed, thawed aliquots were grown on Columbia Agar with 5% Sheep Blood (Biomérieux, France).

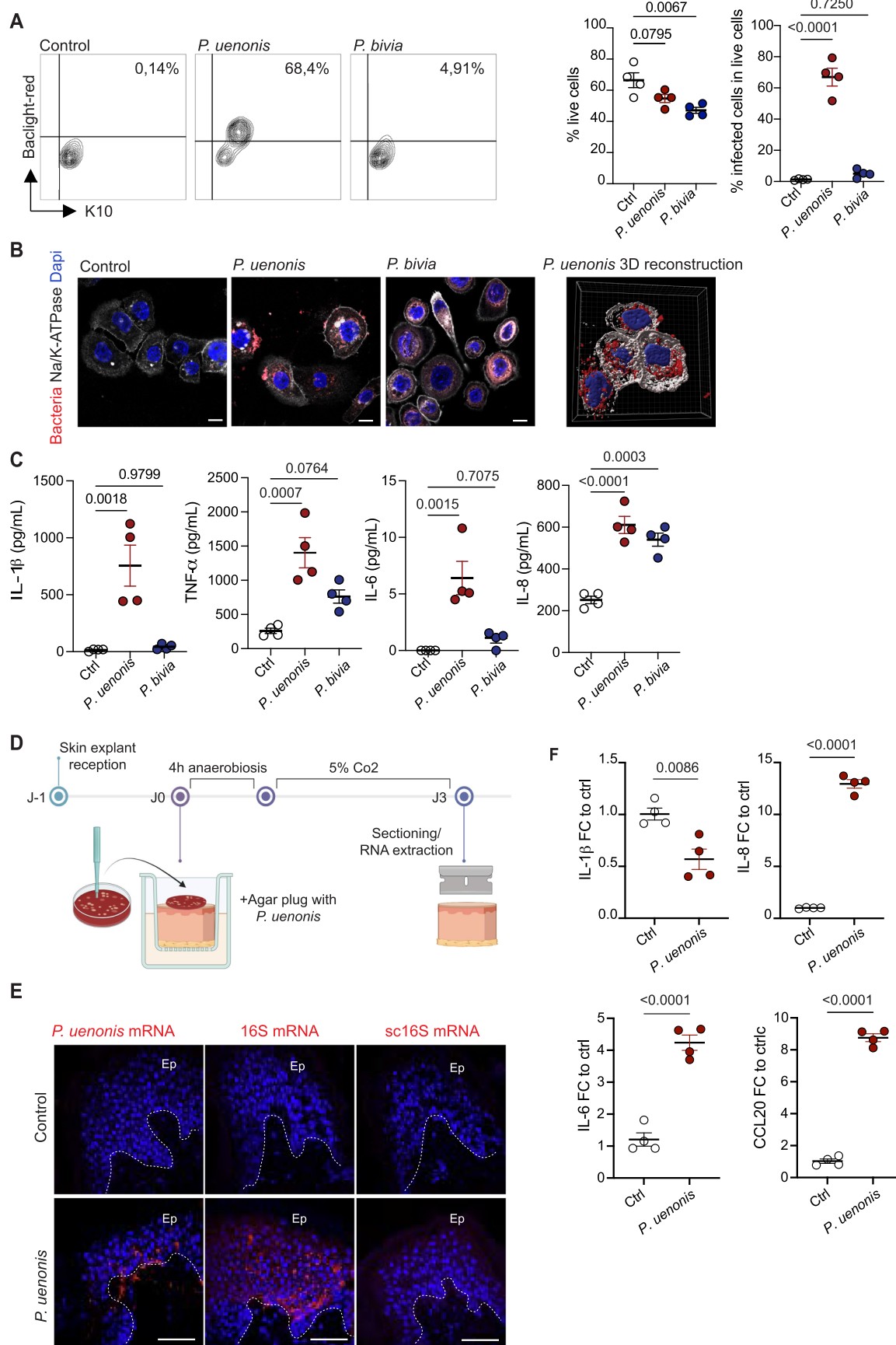

◄ **Figure 5.** *Porphyromonas* infects keratinocytes and triggers inflammation.

KCs isolated from four healthy donors were exposed to fluorescently labeled *P. uenonis*, *P. bivia* or culture medium as control (Ctrl), for 48 h under anaerobic conditions. (A) Representative flow cytometry profiles of bacteria-exposed K10-labeled KCs are shown (left) with comparison of mean % of live and infected cells (right). (B) Confocal images of KCs, delimited by the membrane marker Na/K-ATPase and stained with Dapi, upon infection with fluorescently labeled bacteria. The right panel shows a 3D reconstruction of KCs infected with *P. uenonis*. Scale bars: 10 μm. (C) Cytokine production by KCs exposed to *P. uenonis*, *P. bivia* or culture medium (Ctrl) for 48 h are compared. (D) Schematic overview of the healthy skin explant invasion methodology. Explants maintained alive in a gel-like matrix were covered with blood-agar plugs that were either sterile (Ctrl) or covered by confluent *P. uenonis* colonies. (E) Representative confocal images of explant sections hybridized with an anti-*P. uenonis* mRNA probe, pan-bacteria 16S mRNA or scrambled 16S mRNA probes as controls. Nuclei were stained in Dapi (blue). Scale bars: 50 μm. (F) Differential expression of IL-1β, IL-8, IL-6, and CCL20 genes as assessed by qPCR, in *P. uenonis*-exposed and Ctrl explants. Data are duplicates measurements from two independent donors. Statistical analyses in (A, C, F) used ordinary one-way ANOVA with Tukey's multiple comparisons test. Source data are available online for this figure.

Anaerobic conditions were maintained using GenBag anaerobic systems (Biomerieux, France). For strain identification, genomic DNA extraction was performed on a 24 h bacterial liquid culture using the DNA Easy UltraClean microbial kit (QIAGEN). Short-read genome sequencing was performed by the P2M Sequencing Platform at the Institut Pasteur (Paris, France) using the Illumina Nextera XT DNA Library Prep Kit and the NextSeq 500 system sequencer. Paired-end (PE) sequencing reads were clipped and trimmed with AlienTrimmer v. 2.0, corrected with Musket v. 1.1 and subjected to a digital normalization procedure with ROCK v. 1.9.3 software. All programs were used with their default settings. The remaining processed reads were assembled and scaffolded with SPAdes v. 3.15.2. Average nucleotide identity (ANI) was calculated between pairs of type strain genomes using FastANIv1.34 with default parameters.

### Bacterial flow cytometry

Igs binding to live bacteria were quantified by flow cytometry as previously described (Moor et al, 2016). In brief, bacteria collected from agar plates were resuspended in 0.2-μm-filtered PBS 2% BSA buffer (BFCM buffer). After washing, bacteria were stained with Zombie Nir viability dye (Biolegend, Cat #423105) at a 1:500 dilution for 15 min at room temperature. After counting using a CytoFlex flow cytometer, bacterial suspensions were aliquoted at $2 \times 10^6$ bacteria per well in 96-well plates. Sera, decomplemented and filtered, were diluted in BFCM buffer and used for bacterial incubation. Negative controls included human serum depleted of IgA/IgM/IgG (Sigma, Cat# S5393-1VL-PW), while specificity controls included IgA/IgM/IgG supplemented with anti-GFP recombinant human antibodies (Bio-Rad, Cat# HCA189, HCA191, and HCA192). Secondary antibodies targeting IgA, IgM, and IgG were used, followed by fixation with 2% paraformaldehyde, before acquisition on FACS Cytoflex (Beckman Coulter, France). Data were analyzed using FlowJo X software, and antibody titers were reported in arbitrary units (AU), as a ratio of median of fluorescence in HC and HS serum sample to median of fluorescence of a standard serum ERM (ERM®- DA470k/IFCC) treated in parallel.

### Anti-bacterial Ig depletion

Bacteria grown on blood-agar plates were resuspended in filtered PBS containing 2% FBS. Pellets were collected after centrifugation (1500 × *g*, 30 min, 4 °C) and the concentration was adjusted to $10^9$ bacterial/mL. Bacteria were incubated for 2 h at 4 °C with an equal volume of decomplemented sera pools. Three independent pools of sera from four different individuals were tested. The quality of Ig depletion was controlled by bacterial flow cytometry (Fig. EV2B).

### ELISAs of anti-bacterial Igs

Anti-bacterial Igs were quantified by ELISA as previously described (Benckert et al, 2011). In brief, bacteria were grown in LB medium from a single colony until the culture reached an OD600nm of 0.1, fixed with 0.2% paraformaldehyde (Sigma) for 30 min, washed and coated onto poly-L-lysine-treated high-binding 96-well ELISA plates. After washing with 0.05% Tween 20-PBS (PBST), the plates were blocked for 2 h with 2% BSA, 1 mM EDTA in PBST (blocking buffer), washed again, and incubated with serum diluted 1:500 in PBS, in triplicate. To quantify sIgA levels, plates were coated overnight with 125 ng/well of mouse anti-human secretory component antibodies (GA-1, Sigma). After washing with PBST, the plates were blocked, washed again and incubated with serum diluted 1:50, followed by three consecutive 1:3 dilutions in PBS, in duplicate. Mouse secretory IgAs were quantified with the Mouse IgA ELISA kit (Assay Genie, Cat# MOES01461), according to the manufacturer's instructions. Polyreactivity ELISA was performed as previously described (Planchais et al, 2022). High-binding 96-well ELISA plates were coated overnight with 500 ng/well of purified double-stranded DNA (dsDNA), KLH, LPS, lysozyme (LZ), thyroglobulin (TB) (Sigma-Aldrich), 250 ng/well of Peptidoglycan from *B. subtilis* (PGN) (Invivogen) and recombinant MAPK14 (Planchais et al, 2019) in PBS. After the blocking and washing steps with 0.001% Tween 20-PBS, the plates were incubated with serum diluted 1:500, followed by three consecutive 1:3 dilutions in PBS, in duplicate. mGO53 (Wardemann et al, 2003) and ED38 (Meffre et al, 2004) were used as negative and polyreactive controls, respectively. The plates were developed by incubating them for 1 h with goat HRP-conjugated anti-human IgG, -IgG1, -IgG2, -IgG3, -IgG4, -IgA, -IgA1, or -IgA2 antibodies (0.8 μg/mL final concentration) and by adding 100 μL of HRP chromogenic substrate (ABTS solution, AAT Bioquest, Cat# 11001) after the PBST washing steps. Antibody references were HRP-goat anti-human IgG, Fcγ secondary antibody (Jackson Immunoresearch; Cat# 109-035-098), mouse anti-human IgG1 Fc secondary antibody, HRP (Thermo Fisher; Cat# A-10648), mouse anti-human IgG2 Fd secondary antibody, HRP (Thermo Fisher; Cat# 05-0520), mouse anti-human IgG3 (Hinge) secondary antibody, HRP (Thermo Fisher; Cat# 05-3620), mouse anti-human IgG4 Fc secondary antibody, HRP (Thermo Fisher; Cat# A-10654), HRP-goat anti-human serum IgA, α chain secondary antibody (Jackson Immunoresearch; Cat# 109-035-011), mouse anti-human IgA1-HRP (SouthernBiotech; Cat# 9130-05), mouse anti-human IgA2-HRP (SouthernBiotech; Cat# 9140-05-). Optical densities were measured

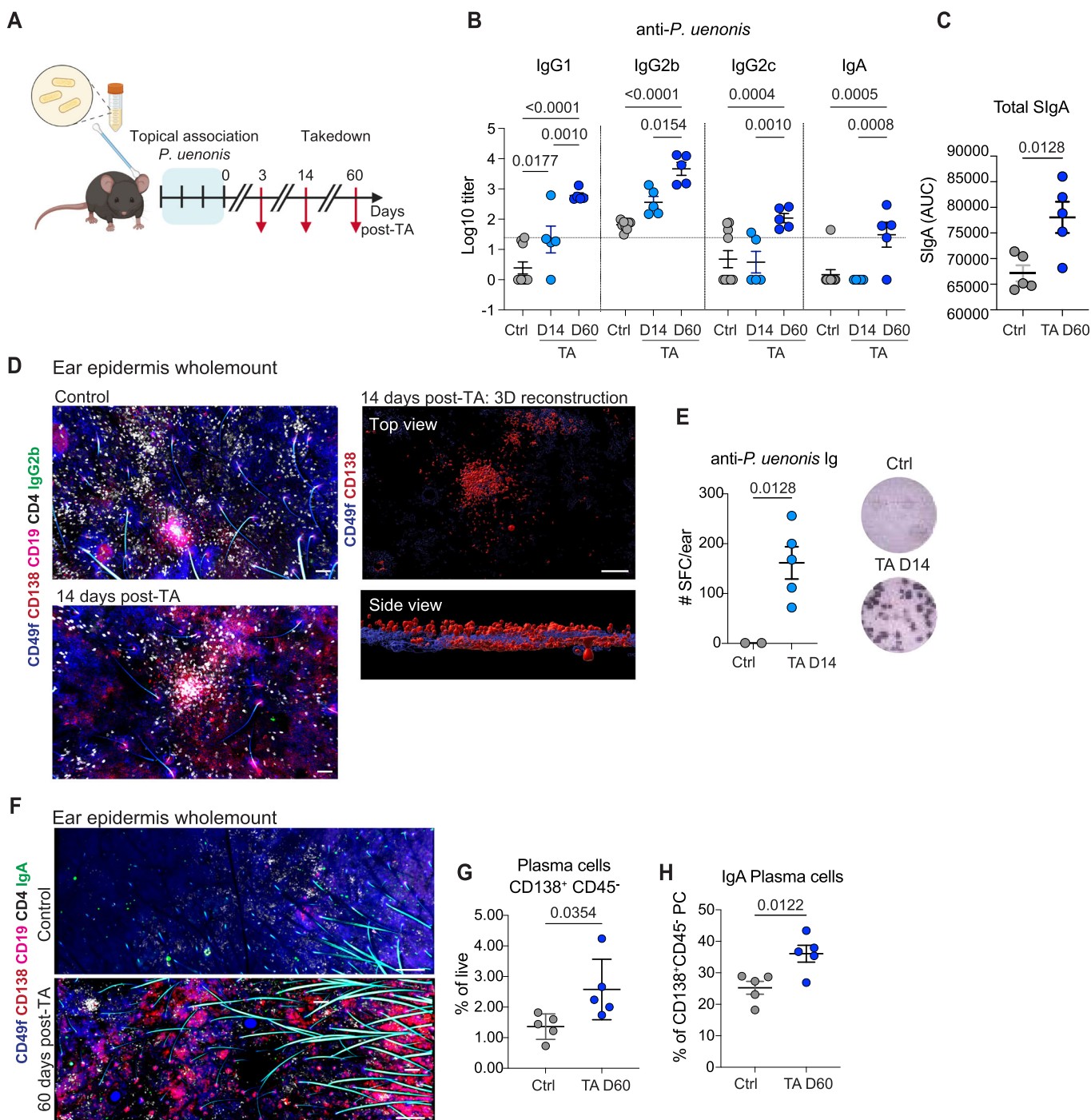

**A** Topical association *P. uenonis* ... Takedown ... Days post-TA

**B** anti-*P. uenonis*

**C** Total SIgA

**D** Ear epidermis wholemount

Control

14 days post-TA

14 days post-TA: 3D reconstruction

Top view

Side view

**CD49f CD138 CD19 CD4 IgG2b**

**CD49f CD138**

**E** anti-*P. uenonis* Ig

Ctrl

TA D14

**F** Ear epidermis wholemount

**CD49f CD138 CD19 CD4 IgA**

Control

60 days post-TA

**G** Plasma cells CD138⁺ CD45⁻

**H** IgA Plasma cells

at 405 nm (OD405 nm), and background values from wells incubated with PBS alone were subtracted. All experiments were performed using a HydroSpeed™ microplate washer and a Sunrise™ microplate absorbance reader (Tecan, Männedorf, Switzerland).

**In situ hybridizations**

ISH was performed using the RNAscope kit (Bio-Techne, Minneapolis, Cat# 323100) following the manufacturer's instructions. Probes for *P. uenonis* (Cat# 838331) and *P. bivia* (Cat# 1241421-C1) were purchased from Bio-Techne. Their hybridization

efficacy and selectivity were determined on live bacterial suspensions (Fig. EV3B). As positive control, we used the UBC probe (Cat# 320861) and as negative control, an anti- probe targeting *DapB* mRNAs of *E. coli* (Cat# 320871). Slides were scanned using the LSM700 Zeiss Confocal microscope, and FIJI software was used for image analysis.

**Immunofluorescence detections in human skin sections**

Tissue sections embedded in OCT were rehydrated in PBS for 10 min and treated with 50 μM NH₄Cl for 15 min at RT to quench

**Figure 6.  Topical association with *P. uenonis* triggers PC recruitment in vivo.**

(A) Experimental setup: Mouse ears were topically associated (TA) with *P. uenonis* for 4 days. Skin and serum samples were collected at 3, 14, or 60 days post-TA. Control mice were treated with broth only. (B) Serum levels of anti-*P. uenonis* IgG1, IgG2b, IgG2c, and IgA were measured in control and TA groups at 14- and 60-days post-TA. Data were mean Log10 Ig titers in each individual mouse ($n = 5$) ± SD. (C) Total secretory IgA (sIgA) levels in serum were assessed 60 days post-TA. Data were the mean area under the curve (AUC) determined with serially diluted sera in each individual mouse ($n = 5$) ± SD. (D) Representative confocal images of mouse ears epidermis wholemount 14 days post-TA. Ears from control and TA mice were stained for CD49f (KCs), CD138 (PCs), CD19 (B cells), CD4 (helper T cells), and IgG2b-expressing cells. Scale bars: 80 μm. The right panel shows 3D renderings of the TA ear region with emphasis on CD138$^+$ cells. (E) ELISpot analysis of mouse ear skin cells assessing PCs producing Igs targeting *P. uenonis*. Dark spots indicate Ig+ cells reactive to *P. uenonis*. Data were mean spot-forming cell numbers per ear in each individual mouse (2 Ctrl, 5 TA) ± SD. Shown blots are from one representative mouse in each group. (F) Representative confocal images of mouse ears 60 days post-TA. Ears from control and TA mice were stained for CD49f (KCs), CD138 (PCs), CD19 (B cells), CD4 (helper T cells), and IgA-expressing cells. Images depict control ears (left panel) and TA ear (right panel). Scale bars: 300 μm. (G) FACS analysis of live PCs (defined as live CD138$^+$ CD45$^-$ cells) in ear skin 60 days post-TA with *P. uenonis*. Data were mean % from each individual mouse ($n = 5$) ± SD. (H) FACS analysis of IgA-expressing PCs in ear skin 60 days post-TA with *P. uenonis*. Data were mean % from each individual mouse ($n = 5$) ± SD. Statistical analyses used two-way ANOVA with Tukey's multiple comparisons test in (B); unpaired *t*-test in (C, E, G, H). Source data are available online for this figure.

**Table 4.  Primers used for qRT-PCR.**

| Gene | Primers |
|------|---------|
| IL-8 | F: GAGAGTGATTGAGAGTGGACCAC |
|      | R: CACAACCCTCTGCACCCAGTTT |
| CCL20 | F: AAGTTGTCTGTGTGCGCAAATCC |
|       | R: CCATTCCAGAAAAGCCACAGTTTT |
| HPRT1 | F: CATTATGCTGAGGATTTGGAAAGG |
|       | R: CTTGAGCACACAGAGGGCTACA |
| IL-6 | F: AGACAGCCACTCACCTCTTCAG |
|      | R: TTCTGCCAGTGCCTCTTTGCTG |
| IL-1β | F: CCTGTCCTGCGTGTTGAAAGA |
|       | R: GGGAACTGGGCAGACTCAAA |

autofluorescence and then blocked with 3% bovine serum albumin (BSA) in PBS for 30 min at room temperature in a humid chamber. Sections were incubated overnight at 4 °C with the following primary antibodies, diluted in 3% BSA in PBS: goat anti-CD138/Syndecan-1 (1:200, Fisher Scientific, Cat# PA5-47395), mouse anti-IgA (1/100, Proteintech, Cat# 60099-1-IG) and rabbit anti-IgG (1/100, Fisher Scientific, Cat# MA5-42729). After incubation, sections were washed three times with PBS for 5 min each. Secondary antibodies were then added and incubated for 1 h at RT: Donkey Anti-Goat Alexa Fluor 555 (1/500, Invitrogen, Cat# A-21432), donkey anti- rabbit Alexa Fluor 647 (1/200, Invitrogen, Cat# A-31573) and Goat anti-mouse Alexa Fluor 488 (1/200, Invitrogen, Cat# A-11001). Nuclei were counterstained with Hoechst (1:5000) for 5 min at RT, followed by three additional PBS washes. Finally, tissue sections were sealed with a coverslip using ProLong Diamond mounting solution (Thermo Fisher Scientific). Images were acquired using a Zeiss LSM700 confocal laser-scanning microscope.

### Human primary KCs

Primary epidermal human KCs were obtained from ATCC (Cat# ATCC-PCS-200-011). All experiments were conducted with cells at passage numbers <4 and found to be negative for mycoplasma contamination. KCs were grown in EpiLife (Invitrogen, Cat# EPI-500-CA), supplemented with EpiLife Human Keratinocyte Growth Supplement (HKGS) (Invitrogen, Cat# S-001-5), penicillin/streptomycin (P/S) (Invitrogen, Cat# 15140-122) and the antimycotic Amphotericin B solution (Sigma, Cat# A2942-20ml). For infection

experiments, KCs were plated in fresh culture media in six-well plates at a density of $10^5$ cells per well and cultured overnight to allow adhesion. Bacteria grown on blood agar in an anaerobic atmosphere, as previously described, have been washed and resuspended in PBS at $10^6$ CFU/ml, then stained with Backlight Red bacterial stain (Thermo Fisher Scientific, Cat# B35001) for 30 min in the dark in anaerobic conditions. Stained bacteria were washed with PBS, then added to the KC in the six-well plates at multiplicities of infection (MOI) of 10. After 48 h of incubation at 37 °C in 5% $CO_2$ (anaerobic conditions), KCs were collected, and the presence of bacteria was confirmed by flow cytometry (CytoFLEX, Beckman Coulter) or confocal imaging. For imaging, cell membranes were stained with Na$^+$/K$^+$ ATPase antibody (1:500, Abcam, Cat# AB76020) at a dilution of 1:500 overnight at 4 °C. Nuclei were counterstained with Hoechst (1:5000) for 5 min at RT. Images were acquired on a Zeiss LSM700 confocal laser-scanning microscope. For three-dimensional (3D) reconstructions, Imaris 64 software (v 9.2.0; Bitplane) was utilized at the Image Analysis Hub (Institut Pasteur). To measure cytokine production in response to infection, KCs were stimulated for 6 h with PMA (1 ng/ml) and ionomycin (500 ng/ml) after 48 h of infection. The presence of cytokines in culture supernatants was assessed by LEGENDplex™, following the manufacturer's instructions.

### Mouse primary KCs

Mouse KCs were derived from mouse pups (P1–P2). Following euthanasia, the mouse pups were swabbed with 70% ethanol, and the skin was collected and immediately placed in cold PBS containing P/S. After removing the subcutaneous tissue with a scalpel, the skin was cut into 3–4 mm wide strips and placed into a solution of dispase (STEMCELL Technologies Inc., Cat# 07913) containing 0.5% P/S. After an overnight incubation at 4 °C, the epidermis was separated from the dermis using tweezers and transferred to cold PBS with P/S. The separated epidermal sheets were cut into 1 mm² pieces using scissors and transferred into a warm solution of Trypsin/EDTA for single-cell suspension (incubation for 5 min at 37 °C with shaking). Digestion was stopped by transferring the cell suspension into DMEM supplemented with 10% FBS and 1% P/S. Debris were removed by filtering the cell suspension through 70 μm and then 40 μm mesh filters, followed by centrifugation at 1200 rpm for 5 min. KC pellets were resuspended in DMEM with 10% FBS and 1% P/S and seeded into cell culture flasks. The following day, the medium was removed and replaced with KC growth medium (PromoCell, Cat# C-20011).

KCs were maintained at 37 °C in an atmosphere of 5% $CO_2$ with humidity and were passaged with 0.25% Trypsin when they reached 70% confluence. Mouse KCs found to be negative for mycoplasma contamination were infected with *P. uenonis* as described with human KCs.

### Human autoantibody detection

For autoantibody detection, serum samples were diluted 1:100 in PBS with 2% FBS, filtered through a 0.22-μm filter, and then incubated with the skin sections of KCs cultures overnight at 4 °C. Skin sections and KCs were washed three times with PBS and incubated with Alexa Fluor 488-conjugated anti-human IgG (Invivogen, France) for 1 h at 37 °C. Nuclei were stained with Hoechst (Invivogen, France). Images were acquired using a fluorescence microscope (Olympus CK53) at 10x magnification and analyzed with FIJI software.

### Human skin explants

Human skin samples (NativeSkin®) were purchased from Genoskin Inc. (Toulouse, France) and obtained anonymously from donors who provided informed written consent. The study protocol was approved by the French Ethical Committee (Comité de Protection des Personnes) and the French Ministry of Research. Human skin samples from abdominal reduction surgery were sourced from two healthy female donors (aged 27 and 46 years) and biostabilized in Genoskin's patented matrix. Upon receipt, the explants were cultured at 37 °C in an atmosphere of 5% $CO_2$ in 12-well plates containing 1 mL of the provided medium, which did not contain antibiotics (Genoskin, Toulouse, France). For the infection, an 8 mm punch biopsy was used to aseptically collect anaerobic blood-agar plugs that were confluent with *P. uenonis* and *P. bivia*. The agar plugs were placed on top of the skin explants, and an additional 1 mL of culture medium without antibiotics was added to each well. The plates were incubated in an anaerobic atmosphere generated using GenBag Anaerobic (Biomerieux, France) at 37 °C and 5% $CO_2$ for 4 h. After this period, the GenBag Anaerobic was removed, and the explants were maintained with the agar plugs for an additional 72 h. Following this incubation, the biopsies were collected and then sectioned in two for RNA extraction and OCT embedding as described earlier.

### Quantitative reverse-transcription PCR

Skin explant biopsies were homogenized using a Precellys® tissue homogenizer (Bertin Technologies) with 2.8 mm ceramic beads (Bertin Technologies, Cat# 000911-LYSK0-A.0). Total RNA was extracted using the RNeasy® Fibrous Tissue Mini Kit (Qiagen, Cat# 74704) according to the manufacturer's instructions. First-strand cDNA was synthesized from 500 ng of total RNA using the High-Capacity cDNA Reverse Transcription Kit (Applied Biosystems, Cat# 4368814). Quantitative PCR (qPCR) was performed using Power SYBR Green PCR Master Mix (Applied Biosystems, Cat# 4367659) and gene-specific primers (see Table 4). The amplification was conducted using a 1 ng cDNA template in a final volume of 10 μL in a 96-well PCR plate. The amplification conditions were as follows: 2 min at 50 °C, 10 min at 95 °C, followed by 40 cycles of 15 s at 95 °C and 1 min at 60 °C on a QuantStudio 3 Real-Time PCR System (Applied Biosystems). The results were normalized to the relative expression of *HPRT*, which served as an endogenous control.

### Mouse TA, tissue processing, and flow cytometry

We used C57BL/6 female mice from the NIAID-Taconic exchange program (mouse strain name Tac 8478): CD45.1 (B6.SJL-Ptprca Pepcb/BoyJ), aged 8 weeks at the start of the experiment. All mice were bred and maintained under pathogen-free conditions at an American Association for the Accreditation of Laboratory Animal Care (AAALAC)-accredited animal facility at the NIAID and housed following the procedures outlined in the Guide for the Care and Use of Laboratory Animals. All experiments were performed at the NIAID under an animal study proposal (LHIM-3E) approved by the NIAID Animal Care and Use Committee. For the TA with bacterial culture, colonies of *P. uenonis* or *P. bivia* were inoculated at an optical density at 600 nm (OD600nm) of 0.01 in Cooked Meat Broth Hemin/Vitamin K and incubated overnight at 37 °C under anaerobic conditions until OD600nm reached 0.8. For the TA with bacteria, each mouse was associated by placing 1 ml of the bacterial suspension (estimated $10^7$ CFUs ml−1) on the entire ear skin surface. Each experimental group included five mice, a standard sample size in the field, sufficient for reproducible phenotypic results. Mice were randomly assigned to treatment groups. Blinding was not performed because treatment groups were housed separately to prevent bacterial transfer between cage mates. Following the TA, the ears were excised and separated into the ventral and dorsal sheets. All enrolled animals were used for analysis; no data points were excluded. Tissue samples were digested in RPMI containing 55 μM β-mercaptoethanol, 20 μM HEPES (HyClone), 0.25 mg/ml Liberase purified enzyme blend (RocheDiagnostic Corp.), 0.2 mg/ml DNase I (Sigma) and incubated for 1.5 h at 37 °C and 5% $CO_2$. Digested skin sheets were homogenized using the Medicon/Medimachine tissue homogenizer system (Becton Dickinson) and spun down at $500 \times g$ for 5 min. To assess bacterial survival in intact mouse skin, swabs and tissue digests were plated on agar under anaerobic conditions, and we performed 16S rDNA Sanger sequencing on the resulting bacterial cultures. Parallel to this, single-cell suspensions were stained with LIVE/DEAD Fixable Blue Dead Cell Stain Kit (Invitrogen) in PBS to exclude dead cells. Cells were stained with the following antibodies: rat anti-mouse CD138-PE (1:100, 281-2, BD Pharmingen, Cat# 561070), rat anti-mouse CD73-Pe-Cy7 (1:400, TY/11.8, Biolegend, Cat# 127223), rat anti-mouse CD273 (1:300, TY25, BD Pharmingen, Cat# 560086), rat anti-mouse CD45-BUV396 (1:300, 30-F11, BD Horizon, Cat# 564279), rat anti-mouse IgD-PerCP-Cy5.5 (1:800, 11-26 c.2a, BioLegend, Cat# 405709), rat anti-mouse/human GL-7-eF450 (GL-7, eBioscience, Cat# 48-5902-82, 1:200), rat anti-mouse B220-PECF594 (1:400, RA3-6B2, BD Horizon, Cat# 562313), rat anti-mouse CD19-BV785 (1:300, 6D5, BioLegend, Cat# 115543). Staining was performed in the presence of FcBlock (Thermo Fisher), 0.2 mg/ml purified rat IgG, and 1 mg/ml of normal mouse serum (Jackson Immunoresearch). Cells were acquired on a BD LSRFortessa cell analyzer (BD Biosciences) equipped with FACSDiva software (v9.0) and analyzed using FlowJo software (v10.8.2).

### Confocal microscopy analysis of mouse tissues

For ear dermis wholemount - ear pinnae were split with forceps, fixed in 1% paraformaldehyde solution (Electron Microscopy Sciences) overnight at 4 °C, and blocked in 1% BSA, 0.25% Triton X blocking buffer for 2 h at room temperature. Tissues were stained overnight at 4 °C in blocking buffer with antibodies: rat anti-mouse

CD19-APC (1/200, 1D3, BD Pharmingen, Cat# 561738), rat anti-mouse CD49f-eF450 (eBioGoH3, eBioscience, Cat# 48-0495-82, 1:200), rat anti-mouse IgG2b-FITC (1/200, R12-3, BD Pharmingen, cat# 553395), rat anti-mouse CD4-AF700 (RM4-5, eBioscience (Thermo Fisher Scientific), Cat# 56-0042-82, 1:200), rat anti-mouse CD138-PE (1:100, 281-2, BD Pharmingen, Cat# 561070), rat anti-mouse IgA-FITC (1:150, C10-3, BD Pharmingen, Cat# 559354). After being washed three times with PBS, tissues were mounted with ProLong Gold (Molecular Probes) antifade reagent. Ear pinnae images were captured on a Leica TCS SP8 confocal microscope equipped with LAS X software. Images were analyzed using Imaris Bitplane software (v10.0).

### ELISAs and ELISPOTs on mouse samples

For the ELISA, mice were bled at sacrifice, and serum samples were analyzed individually in flat-bottom 96-well MaxiSorp microtiter plates (Nunc). Wells were coated with heat-killed bacterial culture in PBS (~10 µg/mL) overnight in PBS at 4 °C. After washing the plates in PBS and blocking with 0.1% bovine serum albumin (BSA)/PBS, samples were serially diluted in 0.1% BSA/PBS, and aliquots were added to corresponding sub-wells. The plates were kept at 4 °C overnight, and after washing in PBS, the plates were incubated with alkaline phosphatase (AP)–conjugated isotype-specific goat anti-mouse antibodies (Southern Biotechnology, Birmingham, AL): goat anti-mouse IgG1-AP (Cat# 1071-04, 1:1000), goat anti-mouse IgA-AP (Cat# 1040-04, 1:1000), goat anti-mouse IgG2b-AP (Cat# 1091-04, 1:1000), goat anti-mouse IgG2c-AP (Cat# 1079-04, 1:1000). Plates were washed, and the phosphatase substrate, p-nitrophenyl phosphatase (NPP) (Sigma-Aldrich), was added to each well. The reaction was read at 405 nm using a BioTek Synergy H1 microplate reader supplied with Gen5 3.14 software. The antibody titers were defined as the interpolated dilutions of the samples, giving rise to an absorbance on the linear part of the curve of 0.4 above the background. For the Elispot assay, Elispot 96-well plates (MultiScreen HTS, Millipore) were coated with heat-killed *P. uenonis* culture in PBS (10 µg/mL). After blocking with 0.1% BSA/PBS, duplicate wells were incubated with mononuclear cells from skin. The cells were incubated for 12 h at 37 °C and 5% $CO_2$. After thorough washing in PBS/0.05%Tween 20, AP-conjugated goat anti-mouse Ig (Southern Biotech, Cat#: 1010-04; 1:1000) in 100 µl per well was added. The bound antibodies marking single-antibody–producing cells were visualized by adding 50 µl per well of Sigmafast BCIP/NBT B5655 1 tablet/10 ml deionized water (Sigma-Aldrich). Spot-forming cells were counted using an ImmunoSpot analyzer (CTL) with CTL Switchboard 2.6.1.

### Statistics

The number of biological and technical replicates, as well as the statistical test used for comparisons, are detailed in each Figure legend.

## Data availability

Metagenomics data have been deposited in NCBI under SRA accession number SUB16009351 and will be made publicly available upon publication.

The source data of this paper are collected in the following database record: biostudies:S-SCDT-10_1038-S44321-026-00407-7.

## Peer review information

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

### The paper explained

#### Problem

Hidradenitis suppurativa (HS) is a chronic, painful, and disabling inflammatory skin disease that affects about 1% of the population. Current treatments, including approved anti-inflammatory biologics, provide only partial and temporary relief for fewer than 60% of patients. The skin of HS patients is often colonized by rare bacterial species, long regarded as harmless bystanders or secondary effects of inflammation. Whether these microbes might play a direct role in disease has remained unknown.

#### Results

Here, we conducted the first immunogenicity study of HS-associated skin bacteria. We found that anti-bacterial antibody responses differed with disease severity. Patients with severe HS showed a distinct immune signature linked to *Porphyromonas uenonis*, a little-studied anaerobic bacterium selectively colonizing HS skin. Strikingly, antibodies targeting *P. uenonis* cross-reacted with skin self-antigens, suggesting a contribution to autoimmunity. Using human skin models, we demonstrated that *P. uenonis* isolated from patients can cross intact skin, invade keratinocytes, and trigger inflammation. However, in immunocompetent mice, the bacteria could not invade the skin, suggesting that HS patient skin has impaired antimicrobial defenses.

#### Impact

Our findings challenge the view that HS-associated bacteria are mere bystanders. Instead, we show that *P. uenonis* may actively drive disease by invading the skin and sustaining chronic inflammation, reminiscent of the role of *Porphyromonas gingivalis* in gum disease. The *P. uenonis* strain we identified may represent a newly evolved skin-invasive variant. We also observed links between other bacteria (e.g., *Staphylococcus lugdunensis*) and milder HS, reinforcing the idea that innate immune defects underlie the failure to control these microbes. Clinically, these results highlight the need to rethink HS treatment: focusing only on suppressing inflammation may inadvertently permit bacterial invasion. Combining targeted antimicrobial strategies with immune-modulating therapies could provide more effective and durable solutions for patients.

transcription factors RORγt and Ahr that leads to IL-17 production by activated B cells. Nat Immunol 14:514–522

Byrd AS, Carmona-Rivera C, O'Neil LJ, Carlucci PM, Cisar C, Rosenberg AZ, Kerns ML, Caffrey JA, Milner SM, Sacks JM et al (2019) Neutrophil extracellular traps, B cells, and type I interferons contribute to immune dysregulation in hidradenitis suppurativa. Sci Transl Med 11:eaav5908

Cai Y, Xue F, Quan C, Qu M, Liu N, Zhang Y, Fleming C, Hu X, Zhang H-G, Weichselbaum R et al (2019) A critical role of the IL-1β-IL-1R signaling pathway in skin inflammation and psoriasis pathogenesis. J Invest Dermatol 139:146–156

Carmona-Rivera C, O'Neil LJ, Patino-Martinez E, Shipman WD, Zhu C, Li Q-Z, Kerns ML, Barnes LA, Caffrey JA, Kang S et al (2022) Autoantibodies present in hidradenitis suppurativa correlate with disease severity and promote the release of proinflammatory cytokines in macrophages. J Invest Dermatol 142:924–935

Chahine AA, Nahhas AF, Braunberger TL, Rambhatla PV, Hamzavi IH (2018) Ertapenem rescue therapy in hidradenitis suppurativa. JAAD Case Rep 4:482–483

Chen YE, Fischbach MA, Belkaid Y (2018) Skin microbiota–host interactions. Nature 553:427–436

Curran AM, Girgis AA, Jang Y, Crawford JD, Thomas MA, Kawalerski R, Coller J, Bingham CO, Na CH, Darrah E (2023) Citrullination modulates antigen processing and presentation by revealing cryptic epitopes in rheumatoid arthritis. Nat Commun 14:1061

Dajnoki Z, Somogyi O, Medgyesi B, Jenei A, Szabó L, Gáspár K, Hendrik Z, Gergely P, Imre D, Póliska S et al (2022) Primary alterations during the development of hidradenitis suppurativa. J Eur Acad Dermatol Venereol 36:462–471

Dang VD, Hilgenberg E, Ries S, Shen P, Fillatreau S (2014) From the regulatory functions of B cells to the identification of cytokine-producing plasma cell subsets. Curr Opin Immunol 28:77–83

Delage M, Jais J-P, Lam T, Guet-Revillet H, Ungeheuer M-N, Consigny P-H, Nassif A, Join-Lambert O (2023) Rifampin-moxifloxacin-metronidazole combination therapy for severe Hurley stage 1 hidradenitis suppurativa: prospective short-term trial and 1-year follow-up in 28 consecutive patients. J Am Acad Dermatol 88:94–100

Egeberg A, Thyssen JP (2023) The optimal biologic treatment target for hidradenitis suppurativa remains undiscovered. Lancet 401:708–710

Enamorado M, Kulalert W, Han S-J, Rao I, Delaleu J, Link VM, Yong D, Smelkinson M, Gil L, Nakajima S et al (2023) Immunity to the microbiota promotes sensory neuron regeneration. Cell 186:607–620.e17

Frew JW, Hawkes JE, Krueger JG (2019) Topical, systemic and biologic therapies in hidradenitis suppurativa: pathogenic insights by examining therapeutic mechanisms. Ther Adv Chronic Dis 10:2040622319830646

Frew JW, Marzano AV, Wolk K, Join-Lambert O, Alavi A, Lowes MA, Piguet V (2021) A systematic review of promising therapeutic targets in hidradenitis suppurativa: a critical evaluation of mechanistic and clinical relevance. J Invest Dermatol 141:316–324.e2

Fritz JH, Rojas OL, Simard N, McCarthy DD, Hapfelmeier S, Rubino S, Robertson SJ, Larijani M, Gosselin J, Ivanov II et al (2011) Acquisition of a multifunctional IgA+ plasma cell phenotype in the gut. Nature 481:199–203

Fyhrquist N, Muirhead G, Prast-Nielsen S, Jeanmougin M, Olah P, Skoog T, Jules-Clement G, Feld M, Barrientos-Somarribas M, Sinkko H et al (2019) Microbe-host interplay in atopic dermatitis and psoriasis. Nat Commun 10:4703

Gribonika I, Band VI, Chi L, Perez-Chaparro PJ, Link VM, Ansaldo E, Oguz C, Bousbaine D, Fischbach MA, Belkaid Y (2025) Skin autonomous antibody production regulates host–microbiota interactions. Nature 638:1043–1053

Gudjonsson JE, Tsoi LC, Ma F, Billi AC, van Straalen KR, Vossen ARJV, van der Zee HH, Harms PW, Wasikowski R, Yee CM et al (2020) Contribution of

plasma cells and B cells to hidradenitis suppurativa pathogenesis. JCI Insight 5:e139930.

Guet-Revillet H, Jais J-P, Ungeheuer M-N, Coignard-Biehler H, Duchatelet S, Delage M, Lam T, Hovnanian A, Lortholary O, Nassif X et al (2017) The microbiological landscape of anaerobic infections in hidradenitis suppurativa: a prospective metagenomic study. Clin Infect Dis 65:282–291

Guilloux C-A, Lamoureux C, Beauruelle C, Héry-Arnaud G (2021) Porphyromonas: a neglected potential key genus in human microbiomes. Anaerobe 68:102230

Hendrickson EL, Xia Q, Wang T, Lamont RJ, Hackett M (2009) Pathway analysis for intracellular Porphyromonas gingivalis using a strain ATCC 33277 specific database. BMC Microbiol 9:185

Hoffman LK, Ghias MH, Cohen SR, Lowes MA (2018a) Polyclonal hyperglobulinaemia and elevated acute-phase reactants in hidradenitis suppurativa. Br J Dermatol 178:e134–e135

Hoffman LK, Tomalin LE, Schultz G, Howell MD, Anandasabapathy N, Alavi A, Suárez-Fariñas M, Lowes MA (2018b) Integrating the skin and blood transcriptomes and serum proteome in hidradenitis suppurativa reveals complement dysregulation and a plasma cell signature. PLoS One 13:e0203672

Hotz C, Boniotto M, Guguin A, Surenaud M, Jean-Louis F, Tisserand P, Ortonne N, Hersant B, Bosc R, Poli F et al (2016) Intrinsic defect in keratinocyte function leads to inflammation in hidradenitis suppurativa. J Invest Dermatol 136:1768–1780

Jiang SW, Whitley MJ, Mariottoni P, Jaleel T, MacLeod AS (2021) Hidradenitis suppurativa: host-microbe and immune pathogenesis underlie important future directions. JID Innov 1:100001

Join-Lambert O, Coignard H, Jais J-P, Guet-Revillet H, Poirée S, Fraitag S, Jullien V, Ribadeau-Dumas F, Thèze J, Le Guern A-S et al (2011) Efficacy of rifampin-moxifloxacin-metronidazole combination therapy in hidradenitis suppurativa. Dermatology 222:49–58

Join-Lambert O, Coignard-Biehler H, Jais J-P, Delage M, Guet-Revillet H, Poirée S, Duchatelet S, Jullien V, Hovnanian A, Lortholary O et al (2016) Efficacy of ertapenem in severe hidradenitis suppurativa: a pilot study in a cohort of 30 consecutive patients. J Antimicrob Chemother 71:513–520

Kimball AB, Jemec GBE, Alavi A, Reguiai Z, Gottlieb AB, Bechara FG, Paul C, Giamarellos Bourboulis EJ, Villani AP, Schwinn A et al (2023) Secukinumab in moderate-to-severe hidradenitis suppurativa (SUNSHINE and SUNRISE): week 16 and week 52 results of two identical, multicentre, randomised, placebo-controlled, double-blind phase 3 trials. Lancet 401:747–761

Lamont RJ, Jenkinson HF (1998) Life below the gum line: pathogenic mechanisms of Porphyromonas gingivalis. Microbiol Mol Biol Rev 62:1244–1263

Lee E-G, Oh JE (2024) From neglect to spotlight: the underappreciated role of B cells in cutaneous inflammatory diseases. Front Immunol 15:1328785.

Lima AL, Karl I, Giner T, Poppe H, Schmidt M, Presser D, Goebeler M, Bauer B (2016) Keratinocytes and neutrophils are important sources of proinflammatory molecules in hidradenitis suppurativa. Br J Dermatol 174:514–521

Lowe MM, Cohen JN, Moss MI, Clancy S, Adler JP, Yates AE, Naik HB, Yadav R, Pauli M, Taylor I et al (2024) Tertiary lymphoid structures sustain cutaneous B cell activity in hidradenitis suppurativa. JCI Insight 9:e169870

Lu J-W, Huang Y-W, Chen T-L (2021) Efficacy and safety of adalimumab in hidradenitis suppurativa: a systematic review and meta-analysis of randomized controlled trials. Medicine 100:e26190

Maboni G, Davenport R, Sessford K, Baiker K, Jensen TK, Blanchard AM, Wattegedera S, Entrican G, Tötemeyer S (2017) A novel 3D skin explant model to study anaerobic bacterial infection. Front Cell Infect Microbiol 7:404

Macchiarella G, Cornacchione V, Cojean C, Riker J, Wang Y, Te H, Ceci M, Gudjonsson JE, Gaulis S, Goetschy JF et al (2023) Disease association of anti–

carboxyethyl lysine autoantibodies in hidradenitis suppurativa. J Invest Dermatol 143:273–283.e12

Meffre E, Schaefer A, Wardemann H, Wilson P, Davis E, Nussenzweig MC (2004) Surrogate light chain expressing human peripheral b cells produce self-reactive antibodies. J Exp Med 199:145–150

Metze D, Jurecka W, Gebhart W, Schmidt J, Mainitz M, Niebauer G (1989) Immunohistochemical demonstration of immunoglobulin A in human sebaceous and sweat glands. J Invest Dermatol 92:13–17

Moor K, Fadlallah J, Toska A, Sterlin D, Balmer ML, Macpherson AJ, Gorochov G, Larsen M, Slack E (2016) Analysis of bacterial-surface-specific antibodies in body fluids using bacterial flow cytometry. Nat Protoc 11:1531–1553

Mulani S, McNish S, Jones D, Shanmugam VK (2018) Prevalence of antinuclear antibodies in hidradenitis suppurativa. Int J Rheum Dis 21:1018–1022

Musilova J, Moran B, Sweeney CM, Malara A, Zaborowski A, Hughes R, Winter DC, Fletcher JM, Kirby B (2020) Enrichment of plasma cells in the peripheral blood and skin of patients with hidradenitis suppurativa. J Invest Dermatol 140:1091–1094.e2

Naik HB, Jo J-H, Paul M, Kong HH (2020) Skin microbiota perturbations are distinct and disease severity-dependent in hidradenitis suppurativa. J Invest Dermatol 140:922–925.e3

Naik HB, Nassif A, Ramesh MS, Schultz G, Piguet V, Alavi A, Lowes MA (2019) Are bacteria infectious pathogens in hidradenitis suppurativa? debate at the symposium for hidradenitis suppurativa advances meeting, November 2017. J Invest Dermatol 139:13–16

Naik S, Bouladoux N, Linehan JL, Han S-J, Harrison OJ, Wilhelm C, Conlan S, Himmelfarb S, Byrd AL, Deming C et al (2015) Commensal–dendritic-cell interaction specifies a unique protective skin immune signature. Nature 520:104–108

Okada T, Konishi H, Ito M, Nagura H, Asai J (1988) Identification of secretory immunoglobulin A in human sweat and sweat glands. J Invest Dermatol 90:648–651

Oliveira CB, Byrd AS, Okoye GA, Kaplan MJ, Carmona-Rivera C (2023) Neutralizing anti–DNase 1 and –DNase 1l3 antibodies impair neutrophil extracellular traps degradation in hidradenitis suppurativa. J Invest Dermatol 143:57–66

Papini M, Simonetti S, Franceschini S, Scaringi L, Binazzi M (1981) Lysozyme distribution in healthy human skin. Arch Dermatol Res 272:167–170

Phan K, Charlton O, Smith SD (2020) Global prevalence of hidradenitis suppurativa and geographical variation—systematic review and meta-analysis. Biomed Dermatol 4:2

Planchais C, Fernández I, Bruel T, De Melo GD, Prot M, Beretta M, Guardado-Calvo P, Dufloo J, Molinos-Albert LM, Backovic M et al (2022) Potent human broadly SARS-CoV-2–neutralizing IgA and IgG antibodies effective against Omicron BA.1 and BA.2. J Exp Med 219:e20220638

Planchais C, Kök A, Kanyavuz A, Lorin V, Bruel T, Guivel-Benhassine F, Rollenske T, Prigent J, Hieu T, Prazuck T et al (2019) HIV-1 envelope recognition by polyreactive and cross-reactive intestinal B cells. Cell Rep 27:572–585.e7

Ring HC, Sigsgaard V, Thorsen J, Fuursted K, Fabricius S, Saunte DM, Jemec GB (2019) The microbiome of tunnels in hidradenitis suppurativa patients. J Eur Acad Dermatol Venereol 33:1775–1780

Ring HC, Thorsen J, Saunte DM, Lilje B, Bay L, Riis PT, Larsen N, Andersen LO, Nielsen HV, Miller IM et al (2017) The follicular skin microbiome in patients with hidradenitis suppurativa and healthy controls. JAMA Dermatol 153:897–905

Riverain-Gillet É, Guet-Revillet H, Jais J-P, Ungeheuer M-N, Duchatelet S, Delage M, Lam T, Hovnanian A, Nassif A, Join-Lambert O (2020) The surface microbiome of clinically unaffected skinfolds in hidradenitis suppurativa: a cross-sectional culture-based and 16S rRNA gene amplicon sequencing study in 60 patients. J Invest Dermatol 140:1847–1855.e6

Ross Y, Ballou S (2022) Association of hidradenitis suppurativa with autoimmune disease and autoantibodies. Rheumatol Adv Pract 6:rkab108

Sabat R, Šimaitė D, Gudjonsson JE, Brembach T-C, Witte K, Krause T, Kokolakis G, Bartnik E, Nikolaou C, Rill N et al (2023) Neutrophilic granulocyte-derived B-cell activating factor supports B cells in skin lesions in hidradenitis suppurativa. J Allergy Clin Immunol 151:1015–1026

Schell SL, Cong Z, Sennett ML, Gettle SL, Longenecker AL, Goldberg SR, Kirby JS, Helm MF, Nelson AM (2023) Keratinocytes and immune cells in the epidermis are key drivers of inflammation in hidradenitis suppurativa providing a rationale for novel topical therapies. Br J Dermatol 188:407–419

Sheridan M, Chowdhury N, Wellslager B, Oleinik N, Kassir MF, Lee HG, Engevik M, Peterson Y, Pandruvada S, Szulc ZM et al (2024) Opportunistic pathogen Porphyromonas gingivalis targets the LC3B-ceramide complex and mediates lethal mitophagy resistance in oral tumors. iScience 27:109860

Somogyi O, Dajnoki Z, Szabó L, Gáspár K, Hendrik Z, Zouboulis CC, Dócs K, Szücs P, Dull K, Törőcsik D et al (2023) New data on the features of skin barrier in hidradenitis suppurativa. Biomedicines 11:127

Świerczewska Z, Lewandowski M, Surowiecka A, Barańska-Rybak W (2022) Microbiome in hidradenitis suppurativa-what we know and where we are heading. Int J Mol Sci 23:11280

Takahashi K, Yanagi T, Kitamura S, Hata H, Imafuku K, Iwami D, Hotta K, Morita K, Shinohara N, Shimizu H (2018) Successful treatment of hidradenitis suppurativa with rituximab for a patient with idiopathic carpotarsal osteolysis and chronic active antibody-mediated rejection. J Dermatol 45:e116–e117

Tohgasaki T, Ozawa N, Yoshino T, Ishiwatari S, Matsukuma S, Yanagi S, Fukuda H (2018) Enolase-1 expression in the stratum corneum is elevated with parakeratosis of atopic dermatitis and disrupts the cellular tight junction barrier in keratinocytes. Int J Cosmet Sci 40:178–186

Vossen ARJV, Ardon CB, van der Zee HH, Lubberts E, Prens EP (2019) The anti-inflammatory potency of biologics targeting tumour necrosis factor-α, interleukin (IL)-17A, IL-12/23 and CD20 in hidradenitis suppurativa: an ex vivo study. Br J Dermatol 181:314–323

Wardemann H, Yurasov S, Schaefer A, Young JW, Meffre E, Nussenzweig MC (2003) Predominant autoantibody production by early human B cell precursors. Science 301:1374–1377

Yilmaz O, Verbeke P, Lamont RJ, Ojcius DM (2006) Intercellular spreading of Porphyromonas gingivalis infection in primary gingival epithelial cells. Infect Immun 74:703–710

Yu W-W, Barrett JNP, Tong J, Lin M-J, Marohn M, Devlin JC, Herrera A, Remark J, Levine J, Liu P-K et al (2024) Skin immune-mesenchymal interplay within tertiary lymphoid structures promotes autoimmune pathogenesis in hidradenitis suppurativa. Immunity 57:2827–2842.e5

## Acknowledgements

We would like to thank Anne Le Flèche-Matéos (Pôle d'Identification Bactérienne of Institut Pasteur), the Mutualized Platform of Microbiology (P2M) and the High-performance Computing Core facility (HPC) of Institut Pasteur for the genome sequencing and their help with bacterial species identification. We acknowledge the ICAReB team for managing the serum collection. We also thank the Agence Nationale de la Recherche for financing the 'Deciphering the causes of hidradenitis suppurativa' project No. ANR-21-CE15-0004-01 (CD, LG-M, and VAA), and Institut Pasteur and INSERM U1224 for core funding of the Immunobiology and Therapy Unit (CD). The research was also supported in part by the Division of Intramural Research, NIAID, NIH (IG, JD, PJP-C, YB).

## Author contributions

**Viviane A Agbogan**: Investigation; Visualization; Methodology; Writing—original draft; Writing—review and editing. **Florence Bugault**: Investigation; Visualization; Methodology; Writing—review and editing. **Laure Guenin-Mace**: Investigation;

Visualization; Methodology; Writing—original draft; Writing—review and editing.
**Inta Gribonika**: Investigation; Visualization; Methodology; Writing—original draft; Writing—review and editing. **Cyril Planchais**: Investigation; Visualization; Methodology; Writing—original draft; Writing—review and editing. **Jean-David Morel**: Investigation; Visualization; Methodology; Writing—review and editing. **Jérémie Delaleu**: Investigation; Visualization; Writing—review and editing. **P Juliana Perez-Chaparro**: Investigation; Writing—review and editing. **Michael Atlan**: Investigation; Writing—review and editing. **Maïa Delage**: Investigation; Writing—review and editing. **Aude Nassif**: Investigation; Writing—review and editing. **Hugo Mouquet**: Supervision; Methodology; Writing—review and editing. **Yasmine Belkaid**: Conceptualization; Supervision; Funding acquisition; Methodology; Writing—review and editing. **Olivier Join-Lambert**: Conceptualization; Funding acquisition; Investigation; Methodology; Writing—review and editing. **Caroline Demangel**: Conceptualization; Supervision; Funding acquisition; Investigation; Visualization; Methodology; Writing—original draft; Project administration; Writing—review and editing.

Source data underlying figure panels in this paper may have individual authorship assigned. Where available, figure panel/source data authorship is listed in the following database record: biostudies:S-SCDT-10_1038-S44321-026-00407-7.

## Disclosure and competing interests statement

The authors declare no competing interests.

# Expanded View Figures

**Figure EV1. Anti-*P. disiens* and *S. lugdunensis* Ig responses and sex effect on anti-bacterial Ig responses.**

(A) Relative level of serum IgA/Gs binding to *P. disiens* and *S. lugdunensis* in HS1 (*n* = 29), HS2 (*n* = 17), HS3 (*n* = 11), or AD patients (*n* = 7), compared to HC (*n* = 32). (B) Heatmap comparing mean levels of anti-bacterial IgAs, IgGs, and IgMs in sera of male (M) or female (F) HS1/2/3 patients to those of HC (left), to each other (middle) or to those of AD patients (right). Statistical analyses used a moderated, FDR-corrected two-tailed *t*-test (limma). *\*p* < 0.05, *\*\*p* < 0.01, *\*\*\*p* < 0.001, and *\*\*\*\*p* < 0.0001.

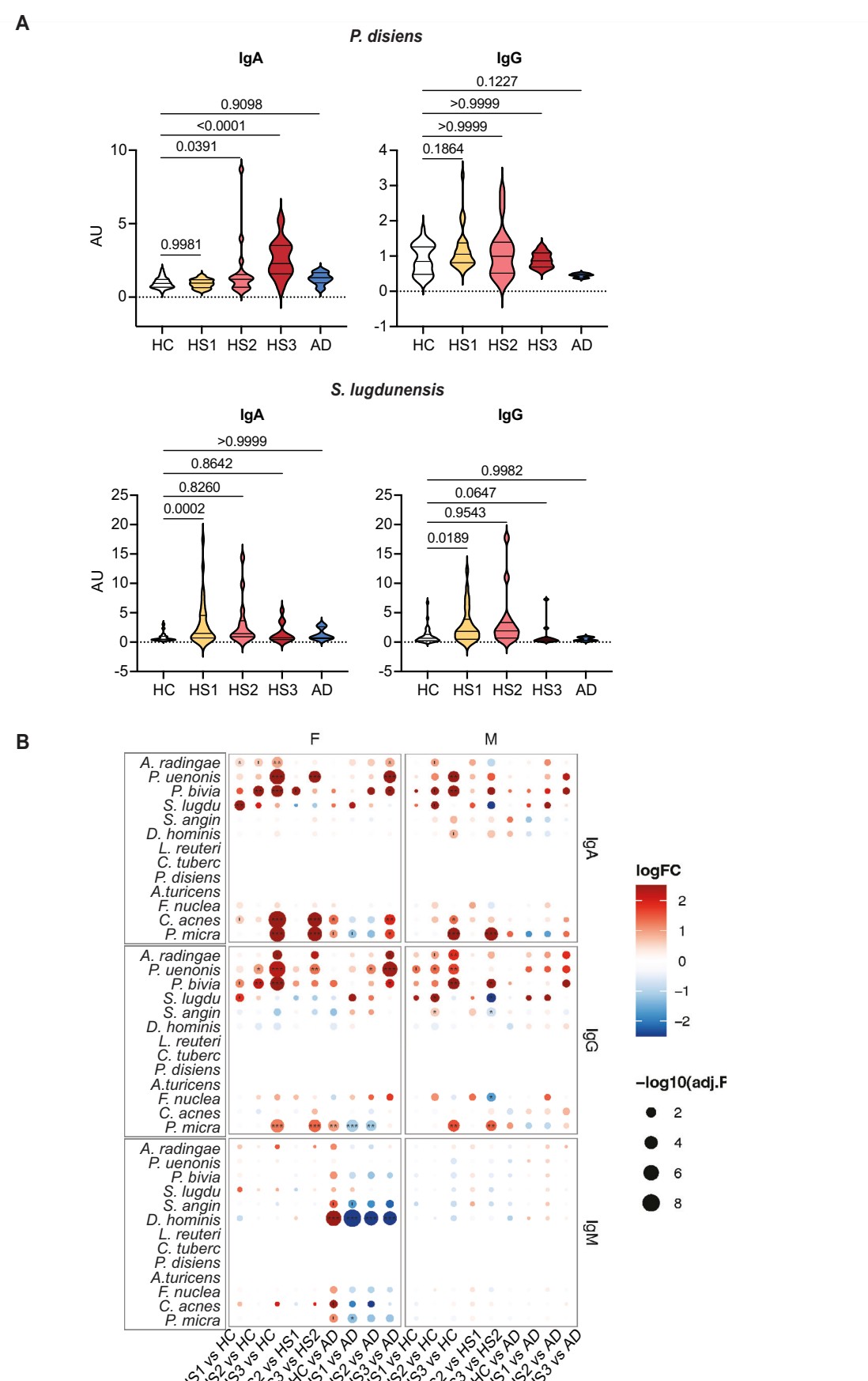

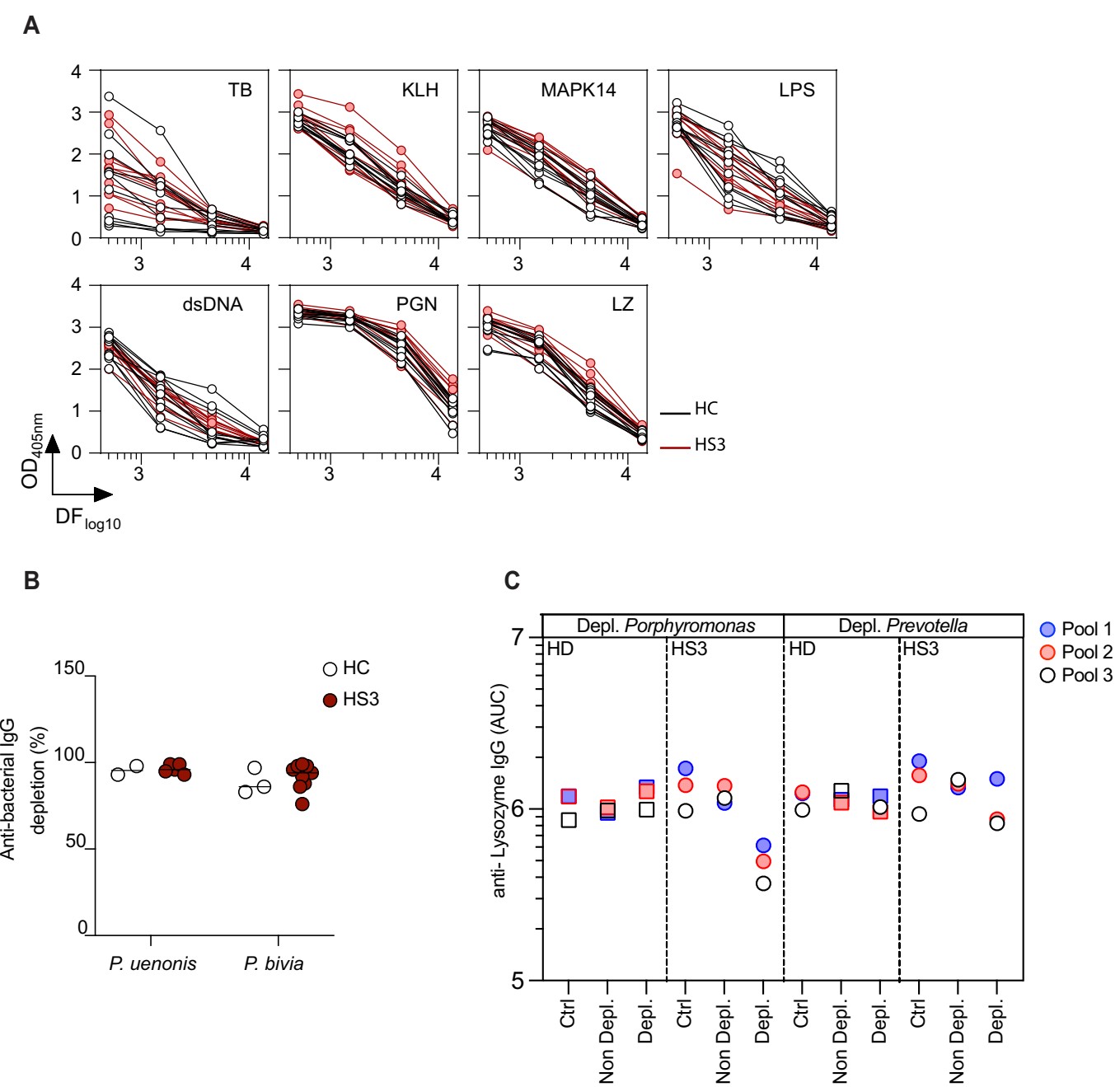

**Figure EV2. IgG autoreactivity and efficacy of anti-bacterial IgG depletion.**

(A) ELISA titration curves of individual HC (black) and HS3 (red) sera for each antigen tested. (B) Loss of anti-bacterial IgG (%), as measured by flow cytometry, in independent pools of HC (n = 2, 3) and HS3 sera (n = 5, 9) sera following depletion on *P. uenonis* or *P. bivia*, respectively. (C) ELISA titration curves of pooled HC (black) and HS3 (red) sera against LZ following depletion on *P. uenonis* or *P. bivia*.

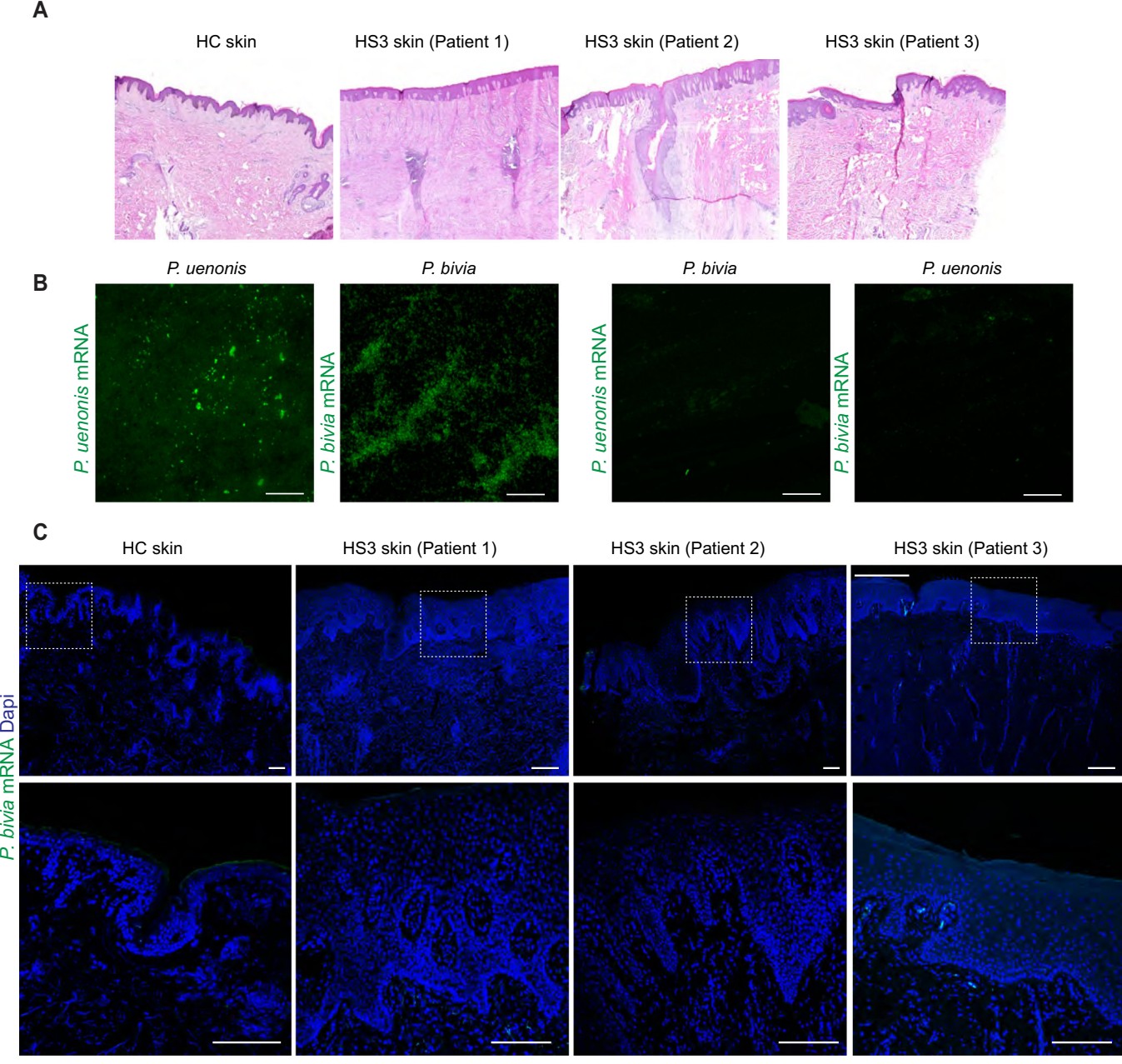

**Figure EV3.** *Prevotella bivia* **does not invade the epidermis in severe HS.**

(**A**) Representative images of skin samples from HC and 3 HS3 patients stained by H&E. (**B**) Validation of mRNA probes targeting *P. uenonis* and *P. bivia* on their respective live bacterial culture (left). Each probe was also tested on the other species (right) as a negative control to assess specificity. (**C**) Representative images of skin samples from HC and 3 HS3 patients hybridized with anti-*P. bivia* mRNA probes, with nuclei counterstained with Dapi. Dotted square indicates the magnified area (bottom panel, Scale bars: 500 μm).

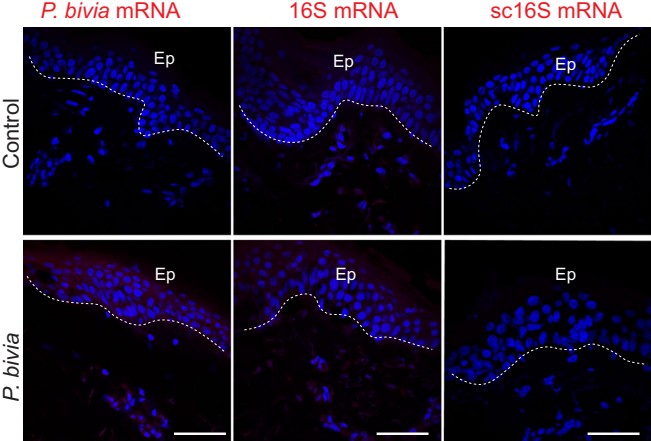

**Figure EV4. Lack of bacterial invasion in *P. bivia*-coated skin explants.**

Representative images of skin explant sections hybridized with an anti-*P. bivia* mRNA probe, pan-bacteria 16S mRNA, or scrambled 16S mRNA probes as controls. Nuclei were stained in DAPI (blue). Scale bars: 50 µm.

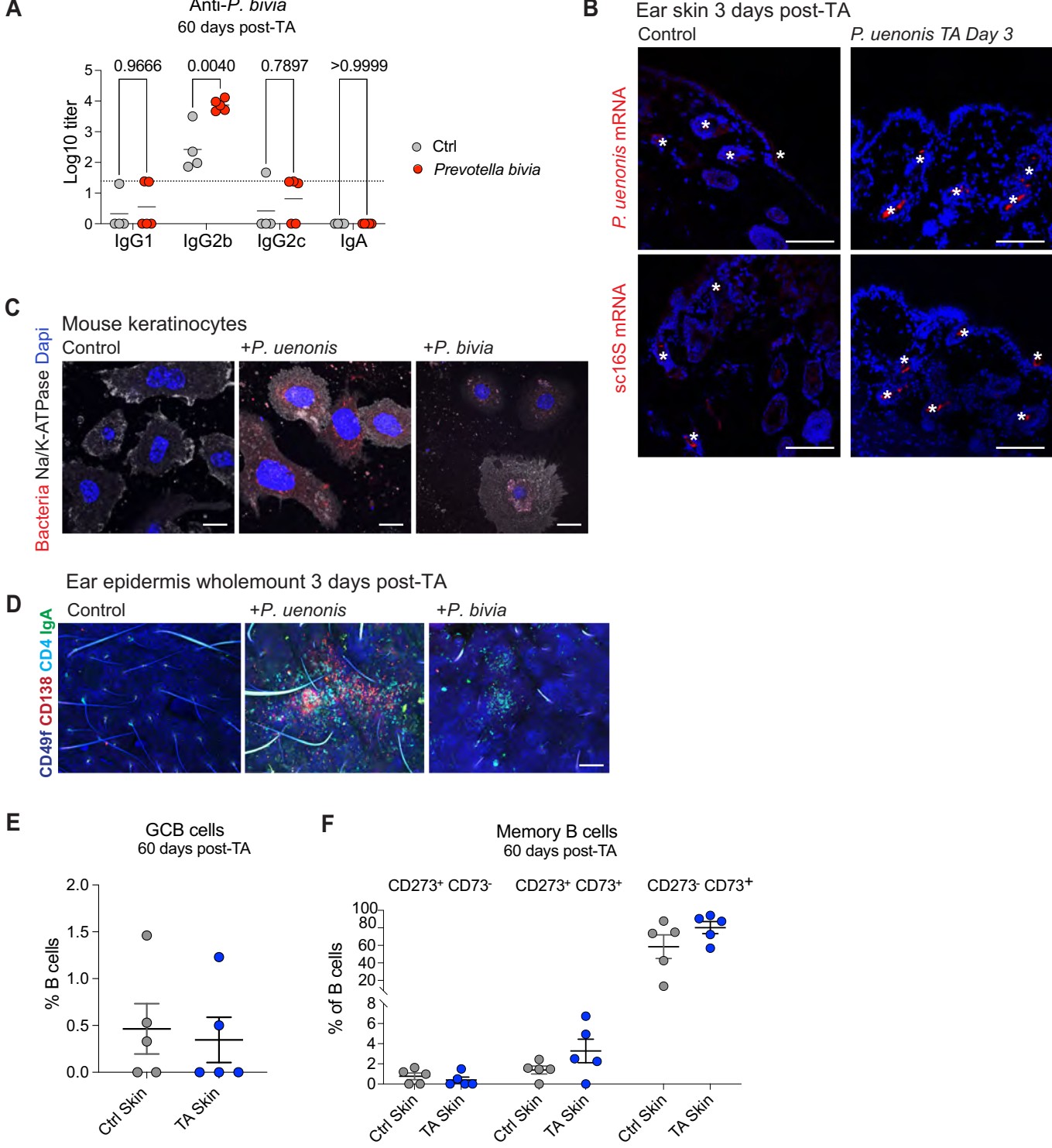

**A** Anti-*P. bivia* 60 days post-TA

**B** Ear skin 3 days post-TA

**C** Mouse keratinocytes

**D** Ear epidermis wholemount 3 days post-TA

**E** GCB cells 60 days post-TA

**F** Memory B cells 60 days post-TA

◄

**Figure EV5.  Topical association with *P. uenonis* triggers PC recruitment in vivo.**

(A) Serum levels of anti-P. *Bivia* IgG1, IgG2b, IgG2c, and IgA were measured in control and TA groups at 60-days post-TA. Each dot represents an individual mouse. Data were mean Log10 Ig titers in each individual mouse ($n = 5$) ± SD. Statistical significance was assessed using two-way ANOVA with Bonferroni's multiple comparisons test. (B) Confocal images of ear skin sections from control and *P. uenonis*-associated mice 3 days post-TA, hybridized with an anti-*P. uenonis* mRNA or scrambled 16S mRNA probe as control (red), with nuclei counterstained with DAPI (blue). White stars indicate non-specific labeling of hair follicles. Scale bars: 100 μm. (C) KCs isolated from wild-type mice were exposed to fluorescently labeled *P. uenonis*, *P. bivia*, or broth (Ctrl) for 48 h under anaerobic conditions. Confocal images show KCs infected with bacteria, Dapi-stained nuclei, and membrane marker Na, K-ATPase. Scale bar: 10 μm. (D) Representative confocal images of mouse ears 3 days post-TA. Control (left), *P. uenonis*-associated (middle), and *P. bivia*-associated (right) mice are shown. Ears were stained for CD49f (KCs), CD138 (PCs), CD4 (helper T cells) and IgA-expressing cells. Scale bars: 80 μm. (E) FACS analysis of germinal center B cells (GCBs) in ear skin 60 days post-TA with *P. uenonis*. Data were mean % in each individual mouse ($n = 5$) ± SD. (F) FACS analysis of memory B cell subsets (defined by their expression of CD273 and/or CD73 markers) in ear skin 60 days post-TA with *P. uenonis*. Data were mean % in each individual mouse ($n = 5$) ± SD.

