## [Peer Review File · EMBO Molecular Medicine]

A skin colonizer disrupts inflammatory and humoral immune defenses in hidradenitis suppurativa

Viviane Agbogon, Florence Bugault, Laure Guenin-Macé, Inta Gribonika, Cyril Planchais, Jean-David Morel, Jérémie Delaleu, Paula Perez-Chaparro, Michael Atlan, Maïa Delage, Aude Nassif, Hugo Mouquet, Yasmine Belkaid, Olivier Join-Lambert, and Caroline Demangel

Corresponding authors: *Caroline Demangel* (demangel@pasteur.fr) , *Olivier Join-Lambert* (olivier.join-lambert@unicaen.fr), *Laure Guenin-Macé* (laure.guenin-mace@pasteur.fr)

Review Timeline:

Submission Date:	24th Jul 25
Editorial Decision:	2nd Sep 25
Revision Received:	7th Oct 25
Editorial Decision:	12th Nov 25
Revision Received:	29th Jan 26
Editorial Decision:	13th Feb 26
Revision Received:	26th Feb 26
Accepted:	3rd Mar 26

Editor: Lise Roth

Transaction Report:

2nd Sep 2025

Dear Prof. Demangel,

Thank you for submitting your manuscript to EMBO Molecular Medicine, and please accept my apologies for the delay in getting back to you, which was due to the annual leave of referees and editorial staff. We have now received feedback from the three reviewers who agreed to evaluate your manuscript. As you will see from the reports below, they acknowledge the novelty, potential medical impact and interest of the study. They nevertheless also raise major concerns including discrepancies between model systems, lack of experimental details and unclear data presentation.

If you feel you can satisfactorily address these points and those listed by the referees, you may wish to submit a revised version of your manuscript. Please attach a covering letter giving details of the way in which you have handled each of the points raised by the referees. A revised manuscript will once again be subject to review, and we cannot guarantee at this stage that the eventual outcome will be favorable.

We are expecting your revised manuscript within three months, if you anticipate any delay, please contact us.

We require:

Additional information on source data and instruction on how to label the files are available

4) A .docx formatted letter INCLUDING the reviewers' reports and your detailed point-by-point responses to their comments. As part of the EMBO Press transparent editorial process, the point-by-point response is part of the Review Process File (RPF), which will be published alongside your paper.

5) A complete author checklist, which you can download from our author guidelines (<https://www.embopress.org/page/journal/17574684/authorguide#submissionofrevisions>). Please insert information in the checklist that is also reflected in the manuscript. The completed author checklist will also be part of the RPF.

6) All Materials and Methods need to be described in the main text using our 'Structured Methods' format. According to this format, the Methods section includes a Reagents and Tools Table (listing key reagents, experimental models, software and relevant equipment and including their sources and relevant identifiers) followed by a Methods and Protocols section describing the methods, ideally using a step-by-step protocol format. The aim is to facilitate adoption of the methodologies across labs. Please download and fill our Reagents and Tools Table template (.docx), which you can find in our author guidelines: <https://www.embopress.org/page/journal/14693178/authorguide#structuredmethods>.

7) Please note that all corresponding authors are required to supply an ORCID ID for their name upon submission of a revised manuscript.

8) It is mandatory to include a 'Data Availability' section after the Materials and Methods. Before submitting your revision, primary datasets produced in this study need to be deposited in an appropriate public database, and the accession numbers and database listed under 'Data Availability'. Please remember to provide a reviewer password if the datasets are not yet public (see <https://www.embopress.org/page/journal/17574684/authorguide#dataavailability>).

In case you have no data that requires deposition in a public database, please state so in this section. Note that the Data

Availability Section is restricted to new primary data that are part of this study.

9) For data quantification: please specify the name of the statistical test used to generate error bars and P values, the number (n) of independent experiments (specify technical or biological replicates) underlying each data point and the test used to calculate p-values in each figure legend. The figure legends should contain a basic description of n, P and the test applied. Graphs must include a description of the bars and the error bars (s.d., s.e.m.). Please provide exact p values.

10) Our journal encourages inclusion of *data citations in the reference list* to directly cite datasets that were re-used and obtained from public databases. Data citations in the article text are distinct from normal bibliographical citations and should directly link to the database records from which the data can be accessed. In the main text, data citations are formatted as follows: "Data ref: Smith et al, 2001" or "Data ref: NCBI Sequence Read Archive PRJNA342805, 2017". In the Reference list, data citations must be labeled with "[DATASET]". A data reference must provide the database name, accession number/identifiers and a resolvable link to the landing page from which the data can be accessed at the end of the reference. Further instructions are available at .

11) We replaced Supplementary Information with Expanded View (EV) Figures and Tables that are collapsible/expandable online. EV Figures should be cited as 'Figure EV1, Figure EV2' etc... in the text and their respective legends should be included in the main text after the legends of regular figures.

12) The paper explained: EMBO Molecular Medicine articles are accompanied by a summary of the articles to emphasize the major findings in the paper and their medical implications for the non-specialist reader. Please provide a draft summary of your article highlighting

13) Author contributions: CRediT has replaced the traditional author contributions section because it offers a systematic machine readable author contributions format that allows for more effective research assessment. Please remove the Authors Contributions from the manuscript and use the free text boxes beneath each contributing author's name in our system to add specific details on the author's contribution. More information is available in our guide to authors.

Please also suggest a visual abstract to illustrate your article as a PNG file 550 px wide x 300-600 px high. A cropped portion of this image will serve as thumbnail for the table of content on our webpage.

16) As part of the EMBO Publications transparent editorial process initiative (see our Editorial at <http://embomolmed.embopress.org/content/2/9/329>), EMBO Molecular Medicine will publish online a Review Process File (RPF) to accompany accepted manuscripts.

In the event of accepted acceptance, this file will be published in conjunction with your paper and will include the anonymous referee reports, your point-by-point response and all pertinent correspondence relating to the manuscript. Let us know whether you agree with the publication of the RPF and as here, if you want to remove or not any figures from it prior to publication. Please note that the Authors checklist will be published at the end of the RPF.

EMBO Molecular Medicine has a "scooping protection" policy, whereby similar findings that are published by others during

review or revision are not a criterion for rejection. Should you decide to submit a revised version, I do ask that you get in touch after three months if you have not completed it, to update us on the status.

I look forward to receiving your revised manuscript.

Yours sincerely,

Lise Roth

**** Reviewer's comments ****

Referee #1 (Remarks for Author):

In this manuscript, the authors explore the role of bacterial dysbiosis in the pathogenesis of Hidradenitis suppurativa (HS), a chronic inflammatory skin disease. The skin of HS patients was previously shown to be colonized by rare bacterial anaerobe species. It is still unclear whether bacterial dysbiosis has a casual link to disease pathogenesis or is a secondary effect. HS is also characterized by a specific autoantibody profile, possibly contributing to autoimmune response.

The authors show that *Porphyromonas uenonis* (Pu) is selectively enriched in the skin of patients with severe HS, which correlates with specific Ig responses. The authors further showed that Pu can cross intact epidermal barrier (using skin explants), infect keratinocyte and induce cytokine expression. In contrast, Pu failed to invade mouse skin in vivo upon topical application, although it has elicited local plasma cell responses. The authors propose that bacteria in HS contribute to disease progression by invading the skin and promoting inflammation. The manuscript is well written and is of high interest.

Specific comments:

- Certain conclusions remain, however, unclear to me or contradictory. Based on the mouse data, the authors suggest that in an immunocompetent host, immune system prevents infection by mobilizing IgA antibodies. However, how does it correlate with the human data shown in Figure 1 and 3: Pu-specific Ig levels increasing with disease severity, while HS3 patients skin samples show Pu invasion? In HS3 patients the humoral response is effectively mounted, yet bacteria persists in epidermis, while in a mouse model Pu does not invade skin. Although the authors showed that Pu is capable of infecting primary mouse keratinocytes, it is still unclear whether it can penetrate intact skin barrier in mice and might reflect species-specific differences. Further B cell responses in mice are not further accompanied by skin inflammation or an autoimmune response, which suggests that the humoral responses are not pathogenic in this context. It seems more likely to me that yet to be identified changes in the skin of HS patients allow bacterial infection of keratinocytes, which leads to (amplification) of inflammatory response. Whether Pu-specific Ig response contributes to disease pathogenesis remains not entirely clear.
- Can author repeat the experiment shown in 2B before and after antibacterial Ig depletion. In figure 2C, there is no differences for the most of the autoantigens tested, only lysozyme, and even then the difference is hard to interpret in the way the data is presented (AUC of the dilution series; it might be helpful to the reader to show the graphs of dilution series as well). Yet, there is a black and white difference in figure 2B. Also in Figure 2D, it is very hard to interpret the data in the way it is presented (what is IgG loss post-depletion -20%?). Can the authors show the data as binding values in the 4 groups for each bacterial species?
- Figure 3D, is there bacterial tissue invasion in patients with HS1 and HS2? How does it correlate with Ig response?
- In a mouse model, do the authors see bacterial invasion into the tissue at an earlier timepoint before the IgA response? If this is not the case, then there are other mechanisms (not IgG or IgA related) that prevent bacterial invasion of the intact epidermis in vivo. If the bacteria is invading the skin at an earlier time-point but is effectively eliminated, can the authors repeat the topical-association model in B cell-deficient mice (also checking skin inflammation)?
- in the discussion section the authors suggest that keratinocytes from HS patients may be more permissive to bacterial invasion. Can the authors further speculate on the mechanisms? Are RNA sequencing data from HS patients available that might give some ideas?

Referee #2 (Comments on Novelty/Model System for Author):

There are some uncertainties about the study cohorts that should be clarified by modifying the tables and the images need to show the pilosebaceous units.

Referee #2 (Remarks for Author):

The manuscript explores a possible pathogenic role of the anaerobic gram-negative rod, *Porphyromonas uenonis* (P.u.), a microbe previously reported to be present in HS skin, particularly in tunnels.

The authors confirm the presence of P.u. in a set of 3 male Hurley stage III patients in comparison to 4 female healthy controls (table 3: skin donors, Figure 1A). Perhaps the origin of this data set is misunderstood by the reviewer?

Comparing antibody titers in sera obtained by Institut Pasteur from 57 HS patients (29 Hurley stage I, 15 stage II, 13 stage III) to 32 healthy control sera obtained by Institut Pasteur and 7 sera from individuals with atopic dermatitis purchased from Tebu-bio, the authors report stage-dependent antibody titers against a spectrum of bacteria, mostly of the IgG and IgA types in HS. Specifically, IgA targeting P.u. is reported to be restricted to stage III HS patients. However, IgG are shown to functionally target keratinocytes. Lysozyme is proposed as an auto-antigen based on a dot blot graph shown in Figure 2C and bacterial pre-absorption data shown in Figure 2D. A non-significant trend towards reduction of IgA after P.u.-depleted sera is taken as evidence that P.u. may have crossed the skin barrier.

The authors also present a series of experiments using human skin biopsies and keratinocytes as well as mouse skin to characterize the ability of P.u. to penetrate epithelia and to elicit pro-inflammatory cytokines.

In the epidermis of skin biopsies obtained from individuals listed in table 3 bacterial P.u. (but not *P. bivia*) RNA is shown in the outermost layers (stratum corneum) in one sample and throughout the epidermis in two samples (Figure 3D). Similar results are shown in skin explants in Figure 5E. Hair follicles or tunnels are not shown in either of these figures. In contrast, in control biopsies and in skin of immunocompetent mice no epidermal invasion of P.u. was shown.

This reviewer recognizes the importance of demonstrating a pathogenic role of individual bacteria in HS, but would like to see more comprehensive evidence to make a sound case for the hypothesis put forward by the authors.

Specific comments

Introduction: The pilosebaceous unit is stated to be the structure where the disease occurs. Apparently the images shown in the manuscript are focused on inter-follicular epidermis. Please expand the images depicted in Figures S3, Figure 2A, Figure 3D, Figure 4, Figure 5E to show the pilosebaceous unit and also show tunnels as P.u. was previously described to be present in tunnels.

Table 1: Please add information by Hurley stage.

Table 3: Please clarify if another table needs to be added to list donors of data shown in Figure 1A?

Results: Please add possible limitations of the data used to demonstrate cross-reactivity.

Results, last paragraph: Do you need to add the modulation of host factors as possible therapeutic strategy?

Methods:

- The study period is mentioned, but not defined. Please provide dates.
- The exclusion criterion progressive skin disease needs clarification.
- Were cryopreserved samples really first fixed in 4% PFA?

Discussion: You mention *Staphylococcus lugdunensis*, and also with regard to Figure 1C and S1: Please discuss how to reconcile previous reports of *Porphyromonas*, *Prevotella*, *Fusobacterium*, *Parvimonas* and others in light of the present data.

Referee #3 (Comments on Novelty/Model System for Author):

I appreciated the use of human tissues/cells, cultured isolates from human disease. High medical impact since this is a novel disease target (dysbiosis) that is treatable.

Referee #3 (Remarks for Author):

In this manuscript, Agbogen et al. investigate the role of the unique microbiome alterations in hidradenitis suppurativa (HS) in pathogenic immune responses. These alterations usually consist of mixed anaerobic bacteria, and here the authors identify *Porphyromonas uenonis* (Pu) as an inducer of IgA/G responses that cross-react with keratinocytes. The cross-reactivity appears to be in part due to lysozyme expressed by KCs. Furthermore, PU effectively penetrates HS and normal skin, supporting its ability to trigger local and systemic humoral immune responses. I congratulate the authors for shedding light on this novel role of anaerobic bacteria in HS, which addresses a longstanding question of whether these altered microbial

populations were cause vs. effect of the disease. I especially appreciated the use of human skin and cultured isolates from HS patients, thus generating findings that have direct clinical relevance. A few suggestions for improvement are below.

- I appreciate the focus on the specific species in Fig. 1A that were pulled out of the metagenomic patient data. However, it would be useful to include a figure with an overview of the metagenomic profiles from the HC, AD, and the HS1/2/3 samples. I think this figure would be helpful to address some questions that arose:
 - o Were the cultured isolates selected for analysis in Fig. 1C also up- or down-modulated in this dataset?
 - o Regarding the observation of IgA/G responses to *S. lugdunensis* in HS1/2 patients, were Staph species like *S. lugdunensis* increased in the metagenomic data from these patients?
- Lysozyme is identified as a cross-reacting antigen to anti-PU antibodies. While statistically significant, the increase is modest in HS3 compared to HC. I wonder if there are additional antigens that could be cross-reacting as well?
- Many bacteria with a peptidoglycan cell wall express lysozyme-like enzymes. Is there anything known about PU or, more generally, *Porphyromonas* lysozymes and whether these might account for the cross-reactivity to KC lysozyme?
- It's interesting that PU has a unique ability to penetrate the epidermis, compared to *P. bivia*. Since *Porphyromonas* species can produce abundant proteases (see DOI: 10.1038/s41522-022-00270-7), is there a potential difference in protease expression between PU and *P. bivia* that explains the ability to penetrate skin?
- In Figure 5C, primary KCs secrete increased IL-1B when infected with PU but not *P. vibria*. However, in the skin explants, IL-1B was decreased with PU treatment. How do you reconcile these opposite results, especially since it doesn't seem to be a donor specific issue?
- Figure 6 - a number of antibodies are used - CD49f, CD138, CD19, etc. Could the authors please indicate what cell types these antibodies are staining, at least in the figure legend?

Referee #1 (Remarks for Author):

In this manuscript, the authors explore the role of bacterial dysbiosis in the pathogenesis of Hidradenitis suppurativa (HS), a chronic inflammatory skin disease. The skin of HS patients was previously shown to be colonized by rare bacterial anaerobe species. It is still unclear whether bacterial dysbiosis has a casual link to disease pathogenesis or is a secondary effect. HS is also characterized by a specific autoantibody profile, possibly contributing to autoimmune response.

The authors show that *Porphyromonas uenonis* (Pu) is selectively enriched in the skin of patients with severe HS, which correlates with specific Ig responses. The authors further showed that Pu can cross intact epidermal barrier (using skin explants), infect keratinocyte and induce cytokine expression. In contrast, Pu failed to invade mouse skin in vivo upon topical application, although it has elicited local plasma cell responses. The authors propose that bacteria in HS contribute to disease progression by invading the skin and promoting inflammation. The manuscript is well written and is of high interest.

We thank the reviewer for the positive evaluation of our work and for recognizing its interest.

Specific comments:

- Certain conclusions remain, however, unclear to me or contradictory. Based on the mouse data, the authors suggest that in an immunocompetent host, immune system prevents infection by mobilizing IgA antibodies. However, how does it correlate with the human data shown in Figure 1 and 3: Pu-specific Ig levels increasing with disease severity, while HS3 patients skin samples show Pu invasion?

In HS3 patients the humoral response is effectively mounted, yet bacteria persists in epidermis, while in a mouse model Pu does not invade skin. Although the authors showed that Pu is capable of infecting primary mouse keratinocytes, it is still unclear whether it can penetrate intact skin barrier in mice and might reflect species-specific differences.

Further B cell responses in mice are not further accompanied by skin inflammation or an autoimmune response, which suggests that the humoral responses are not pathogenic in this context. It seems more likely to me that yet to be identified changes in the skin of HS patients allow bacterial infection of keratinocytes, which leads to (amplification) of inflammatory response. Whether Pu-specific Ig response contributes to disease pathogenesis remains not entirely clear.

We apologize for the lack of clarity in our original statements regarding the role of humoral responses in controlling *P. uenonis* skin invasion in mice and we have clarified our interpretation in abstract, p.6, p.11 and p.27. Our conclusion summarized p. 6 aligns with the Reviewer's: " These findings suggest that in HS, the ability of *P. uenonis* to penetrate the skin, colonize KCs, and elicit inflammatory and humoral responses may play a central role in the development of epidermal immunopathology." We hope that our responses to the specific comments below satisfactorily address the Reviewer's other concerns.

- Can author repeat the experiment shown in 2B before and after antibacterial Ig depletion. We were able to perform this experiment with only two bacterially depleted serum pools, owing to the limited availability of HS3 patient samples. As shown below, depletion on *P. uenonis* or *P. bivia* did not induce a clear reduction in HS3 sera reactivity in this assay.

Although repeating the experiment with a larger number of sera will be necessary to draw definitive conclusions, the current results are consistent with a limited contribution of antibacterial IgGs to LZ binding (Fig. 2D). These data suggest that, while cross-reactivity of antibacterial IgGs directly contributes to autoreactivity in HS skin, additional mechanisms underlie the link between *P. uenonis* infection and autoreactivity. This point and other mechanisms potentially accounting for autoreactivity are now discussed p. 14.

In figure 2C, there is no differences for the most of the autoantigens tested, only lysozyme, and even then the difference is hard to interpret in the way the data is presented (AUC of the dilution series; it might be helpful to the reader to show the graphs of dilution series as well). We used AUC values from ELISA titration curves of serially-diluted antibodies to compare reactivities, as this approach offers a comprehensive and unbiased quantification of antibody binding across the entire concentration range, unlike single-point measurements. As suggested by the reviewer, we now present the raw ELISA titration curves for all tested antigens in Fig. S2A.

We did not detect notable autoreactivity against the autoantigens tested, except for LZ; however, it is likely that additional autoantigens are involved that remain to be identified. See our response to Referee #3 (p. 9), where we propose working hypotheses and approaches to identify them.

Yet, there is a black and white difference in figure 2B. Also in Figure 2D, it is very hard to interpret the data in the way it is presented (what is IgG loss post-depletion -20%?). Can the authors show the data as binding values in the 4 groups for each bacterial species? Binding values are now provided in Fig. S2C to better explain how data in Fig. 2D were obtained, and the corresponding text (p. 8) has been expanded to improve clarity.

- Figure 3D, is there bacterial tissue invasion in patients with HS1 and HS2? How does it correlate with Ig response?

We did not collect skin specimens from HS1 and HS2 patients in the present study and therefore cannot address this question directly. However, their skin microbiota was profiled in parallel with antibacterial Ig responses. The relative abundance of *P. uenonis* was comparable in HS2 and HS3 patients, and higher than in HS1 patients (Fig. 1A). Yet, only HS3 patients displayed *P. uenonis*-reactive Ig responses. As discussed on p. 13, “Interestingly, although mean abundances of *P. uenonis* in the skin microbiota were similar in HS2 and HS3 patients, only HS3 patients developed anti-*P. uenonis* Ig responses (Fig. 1A–C). This discrepancy may reflect the emergence of more virulent strains of *P. uenonis* in HS3, and/or specific defects in barrier defense or bacterial clearance in these individuals.”

- In a mouse model, do the authors see bacterial invasion into the tissue at an earlier timepoint before the IgA response?

Fig. 6B shows that IgA responses developed between D14 and D60 post-TA. As indicated in the Results (p. 10), bacterial invasion was only assessed at D3 and D60, with no signal detected. Since murine KCs are susceptible to *P. uenonis ex vivo* and humoral responses are elicited *in vivo*, we consider it likely that infection occurred at sites or time points not captured in our assay. We now note this limitation of our experimental setup in Results p. 10.

If this is not the case, then there are other mechanisms (not IgG or IgA related) that prevent bacterial invasion of the intact epidermis *in vivo*. If the bacteria is invading the skin at an earlier time-point but is effectively eliminated, can the authors repeat the topical-association model in B cell-deficient mice (also checking skin inflammation)?

Parallel TA of wild-type and B cell-deficient mice with *P. uenonis* were performed to assess the potential protective role of antibodies (see below, D3 post-TA). Because no epidermal colonization was observed in wild-type mice at the time points examined, we opted not to present the B cell-deficient data in the present study.

- in the discussion section the authors suggest that keratinocytes from HS patients may be more permissive to bacterial invasion. Can the authors further speculate on the mechanisms? Are RNA sequencing data from HS patients available that might give some ideas?

Possible mechanisms include a reduced ability to clear microbial invaders due to impaired production of antimicrobial peptides. Supporting this, Hotz et al. reported that KCs from HS patients exhibit defective basal hBD-1 expression and a diminished capacity to induce S100A7/A8 upon ex vivo stimulation (Hotz *et al*, 2016). Consistent with a potential defect in clearing intracellular bacteria, single-cell RNA-seq revealed substantial heterogeneity among KCs in excised HS skin, with only 1 out of the 13 KC subsets expressing antimicrobial genes (Gudjonsson *et al*, 2020). By analogy with *P. gingivalis*, which invades host cells via fimbrial interactions with multiple membrane receptors, it is also plausible that HS-derived KCs may be more susceptible to bacterial infection due to increased expression of such receptors.

Referee #2 (Comments on Novelty/Model System for Author):

There are some uncertainties about the study cohorts that should be clarified by modifying the tables and the images need to show the pilosebaceous units.

Referee #2 (Remarks for Author):

The manuscript explores a possible pathogenic role of the anaerobic gram-negative rod, *Porphyromonas uenonis* (P.u.), a microbe previously reported to be present in HS skin, particularly in tunnels.

The authors confirm the presence of P.u. in a set of 3 male Hurley stage III patients in comparison to 4 female healthy controls (table 3: skin donors, Figure 1A). Perhaps the origin of this data set is misunderstood by the reviewer?

The skin donors listed in Table 3 (HS3 patients and HC) are distinct from the serum donors listed in Table 1 and analyzed in Fig. 1A. To clarify, we have modified the sentence on page 8 referring to skin donors as follows: “To test this hypothesis, we collected skin samples from three additional HS3 patients and HC (Table 3).”

Comparing antibody titers in sera obtained by Institut Pasteur from 57 HS patients (29 Hurley stage I, 15 stage II, 13 stage III) to 32 healthy control sera obtained by Institut Pasteur and 7 sera from individuals with atopic dermatitis purchased from Tebu-bio, the authors report stage-dependent antibody titers against a spectrum of bacteria, mostly of the IgG and IgA types in HS. Specifically, IgA targeting P.u. is reported to be restricted to stage III HS patients. However, IgG are shown to functionally target keratinocytes. Lysozyme is proposed as an auto-antigen based on a dot blot graph shown in Figure 2C and bacterial pre-absorption data shown in Figure 2D. A non-significant trend towards reduction of IgA after P.u.-depleted sera is taken as evidence that P.u. may have crossed the skin barrier.

The trend toward reduced secretory IgA after depletion with *P. uenonis* was not statistically significant. Nevertheless, in the context of strong IgA/IgG responses against *P. uenonis* (Fig. 1) and the altered IgA1 versus IgA2 subclass profile, it provided a rationale for further investigation of potential bacterial invasion of the skin. We have clarified this reasoning in the revised manuscript (p. 8).

The authors also present a series of experiments using human skin biopsies and keratinocytes as well as mouse skin to characterize the ability of P.u. to penetrate epithelia and to elicit pro-inflammatory cytokines. In the epidermis of skin biopsies obtained from individuals listed in table 3 bacterial P.u. (but not *P. bivia*) RNA is shown in the outermost layers (stratum corneum) in one sample and throughout the epidermis in two samples (Figure 3D). Similar results are shown in skin explants in Figure 5E. Hair follicles or tunnels are not shown in either of these figures. In contrast, in control biopsies and in skin of immunocompetent mice no epidermal invasion of P.u. was shown.

Clarifications on the potential role of hair follicles or tunnels in bacterial invasion are provided in our answers to the specific comment below.

This reviewer recognizes the importance of demonstrating a pathogenic role of individual bacteria in HS, but would like to see more comprehensive evidence to make a sound case for the hypothesis put forward by the authors.

We thank the reviewer for acknowledging the novelty of our findings and hope that the responses below satisfactorily address the concerns raised.

Specific comments

Introduction: The pilosebaceous unit is stated to be the structure where the disease occurs. Apparently the images shown in the manuscript are focused on inter-follicular epidermis. Please expand the images depicted in Figures S3, Figure 2A, Figure 3D, Figure 4, Figure 5E to show the pilosebaceous unit and also show tunnels as P.u. was previously described to be present in tunnels.

The tissue sections presented in our manuscript were not chosen to emphasize the interfollicular epidermis, but rather because they displayed *P. uenonis* signals. In light of the increased relative abundance of *Porphyromonas spp.* in both early and chronic HS lesions, as well as in HS tunnels, we considered the possibility that *P. uenonis* may enter the skin preferentially via hair follicles and/or keratinized tunnels. However, although bacterial signals were frequently observed in sections of hair follicles, they were not more abundant than in the adjacent interfollicular epidermis. When tunnels were captured in our HS3 skin sections, we did not consistently detect *P. uenonis* signals along their walls. We believe that further analyses, such as examination of serial adjacent sections and 3D reconstruction of infectious foci in HS3 patient skin and human skin explants coated with the bacteria, would be necessary to rigorously test the hypothesis of a preferential entry via these structures. However, we considered that this work was beyond the scope of the present study.

Table 1: Please add information by Hurley stage.

Corrected p.40

Table 3: Please clarify if another table needs to be added to list donors of data shown in Figure 1A?

All serum donors analyzed in Fig. 1A are listed in Table 1.

Results: Please add possible limitations of the data used to demonstrate cross-reactivity. Although highly significant, depletion of HS3 sera on *P. uenonis* only partially reduced autoreactivity to LZ (Fig. 2D). This suggests that Igs generated during *P. uenonis* infection contribute to autoreactivity through mechanisms other than cross-reactivity—a point now discussed p. 14. We also note that the increased reactivity of HS3 sera to LZ compared to HC (Fig. 2C) was modest. Given the marked reactivity of HS3 sera to human epidermis, this supports recognition of additional autoantigens beyond LZ. In our response to Reviewer 3's specific comment regarding LZ, we outline methods and working hypotheses to identify them. This point is now discussed p. 14.

Results, last paragraph: Do you need to add the modulation of host factors as possible therapeutic strategy?

We chose to introduce this idea in the discussion (p. 15), to contextualize it within the broader implications of our findings : “Our study argues for targeted antimicrobial approaches as adjuncts to immunomodulation, with the potential to synergize with anti-inflammatory biologics and reduce the risk of microbial-driven flares.”

Methods:

-The study period is mentioned, but not defined. Please provide dates.

Corrected p. 15.

-The exclusion criterion progressive skin disease needs clarification.

Corrected p.15.

-Were cryopreserved samples really first fixed in 4% PFA?

Yes. For in situ hybridization, this was the procedure recommended by the manufacturer (Biotechne, Minneapolis, Cat# 323100), and it is fully compatible with immunofluorescence analyses.

Discussion: You mention *Staph lugdunensis*, and also with regard to Figure 1C and S1: Please discuss how to reconcile previous reports of *porphyromonas*, *prevotella*, *fusobacterium*, *parvimonas* and others in light of the present data.

For the IgA/G responses against *S. lugdunensis* in HS1/2 patients, and against anaerobic species (*Porphyromonas*, *Prevotella*, *Fusobacterium*, and *Parvimonas* in particular) in HS3 patients, the induction of Ig responses matches well with the metagenomic profiles of the skin microbiota (Guet-Revillet *et al*, 2014). In other words, the IgA responses reflect the expansion of these species at each stage of severity in our cohort. See also our replies to the first three points from Referee #3.

Referee #3 (Comments on Novelty/Model System for Author):

I appreciated the use of human tissues/cells, cultured isolates from human disease. High medical impact since this is a novel disease target (dysbiosis) that is treatable.

Referee #3 (Remarks for Author):

In this manuscript, Agbogan et al. investigate the role of the unique microbiome alterations in hidradenitis suppurativa (HS) in pathogenic immune responses. These alterations usually consist of mixed anaerobic bacteria, and here the authors identify *Poryphyromonas uenonis* (Pu) as an inducer of IgA/G responses that cross-react with keratinocytes. The cross-reactivity

appears to be in part due to lysozyme expressed by KCs. Furthermore, PU effectively penetrates HS and normal skin, supporting its ability to trigger local and systemic humoral immune responses. I congratulate the authors for shedding light on this novel role of anaerobic bacteria in HS, which addresses a longstanding question of whether these altered microbial populations were cause vs. effect of the disease. I especially appreciated the use of human skin and cultured isolates from HS patients, thus generating findings that have direct clinical relevance. A few suggestions for improvement are below.

We thank the Reviewer for their positive evaluation and interesting suggestions, which have improved the manuscript.

- I appreciate the focus on the specific species in Fig. 1A that were pulled out of the metagenomic patient data. However, it would be useful to include a figure with an overview of the metagenomic profiles from the HC, AD, and the HS1/2/3 samples. I think this figure would be helpful to address some questions that arose:

The metagenomic profiles of lesional skin from the 57 HS patients in this study were previously analyzed as part of an extended cohort of 65 patients, where they were compared with matched unaffected skinfolds (Guet-Revillet *et al*, 2017). The relative abundances shown in Fig. 1A correspond to lesional skin samples from HS patients included in the serological study. This clarification has now been added to the Methods section (p. 16). Skin microbiota samples were collected only from HS patients; therefore, corresponding data are not available for the HC and AD groups.

- o Were the cultured isolates selected for analysis in Fig. 1C also up- or down-modulated in this dataset?

The bacterial isolates used for serological profiling were obtained from additional HS3 patients, for whom we do not have complete metagenomic data and therefore cannot provide their relative abundance in the skin microbiota.

- o Regarding the observation of IgA/G responses to *S. lugdunensis* in HS1/2 patients, were Staph species like *S. lugdunensis* increased in the metagenomic data from these patients?

The relative abundances of *S. lugdunensis* for the HS patients included in the serological study are now show in Fig. 1A. They are consistent with the microbiota analyses of (Guet-Revillet *et al*, 2017), describing *S. lugdunensis* as the predominant species in 22% of HS1 lesions whereas HS3 lesions were mainly characterized by a polymorphic anaerobic microbiota, and HS2 lesions exhibited an intermediate profile.

- Lysozyme is identified as a cross-reacting antigen to anti-PU antibodies. While statistically significant, the increase is modest in HS3 compared to HC. I wonder if there are additional antigens that could be cross-reacting as well?

The increased reactivity of HS3 sera to LZ compared to HC (Fig. 2C) was modest relative to their marked reactivity to human epidermis, suggesting recognition of additional

autoantigens beyond LZ. Identifying these antigens will require immunoprecipitation followed by proteomic analysis, work we plan to pursue in the future, but consider beyond the scope of the present study. Alongside this untargeted approach, a particularly promising hypothesis to test is the cross-reactivity with citrullinated proteins. Indeed, autoantibodies against citrullinated antigens have been detected in HS patient sera (Byrd *et al*, 2019), and both the expression and enzymatic activity of Peptidylarginine Deiminases (PADs, the enzymes mediating citrullination) were elevated in HS skin compared to HC. Intriguingly, *Porphyromonas gingivalis* is the only known prokaryote that encodes a PAD enzyme. The bacterial PAD was shown to citrullinate both the bacterial and human α -enolase, which is abundantly expressed by KCs (Tohgasaki *et al*, 2018). In addition, citrullination was recently shown to stimulate T cells and disrupt immune tolerance in rheumatoid arthritis (RA) (Curran *et al*, 2023). Assessing whether clinical isolates of *P. uenonis* display PAD activity could clarify whether bacteria-driven citrullination of host KC proteins may contribute, directly and/or indirectly, to the development of autoreactivity in HS. The discussion of the revised manuscript has been expanded p. 14 to propose this hypothesis.

- Many bacteria with a peptidoglycan cell wall express lysozyme-like enzymes. Is there anything known about PU or, more generally, *Porphyromonas* lysozymes and whether these might account for the cross-reactivity to KC lysozyme?

This is a very interesting hypothesis. However, we did not find any reports in the literature describing lysozyme-like enzymes in *Porphyromonas* species, nor did we identify enzymes annotated as such in the *P. uenonis* proteome. Sequence comparisons between human lysozyme C (expressed in the epidermis) and *Porphyromonas* enzymes annotated as lytic did not reveal meaningful homologies. Therefore, the cross-reactivity between anti-*P. uenonis* antibodies and host lysozyme is unlikely to derive from recognition of a bacterial lysozyme.

- It's interesting that PU has a unique ability to penetrate the epidermis, compared to *P. bivia*. Since *Porphyromonas* species can produce abundant proteases (see DOI: 10.1038/s41522-022-00270-7), is there a potential difference in protease expression between PU and *P. bivia* that explains the ability to penetrate skin?

This study demonstrates that vaginal isolates of *P. asaccharolytica* and *P. uenonis* secrete metalloproteinases capable of degrading host tissue barriers, similar to *Prevotella* spp. (K *et al*, 2024). Side-by-side comparisons will be needed to assess whether our *P. uenonis* isolate exhibits stronger proteolytic activity. Based on our KC infection data, we propose in discussion that our isolate, unlike *P. bivia*, may share with *P. gingivalis* the ability to invade host cells (de Jongh *et al*, 2023). Together, these findings suggest that *P. uenonis* may cross the skin barrier through both protease-mediated tissue degradation and direct host–cell interactions.

- In Figure 5C, primary KCs secrete increased IL-1B when infected with PU but not *P. bivia*.

However, in the skin explants, IL-1B was decreased with PU treatment. How do you reconcile these opposite results, especially since it doesn't seem to be a donor specific issue?

IL-1 β production in skin explants coated with *P. uenonis* was assessed after 3 days. Previous kinetic studies in mice topically exposed to a contact sensitizer have shown that IL-1 β induction is a rapid and transient phenomenon, declining after 24 h (Matsushima & Takashima, 2009). These observations suggest that, in skin explants exposed to *P. uenonis* for 3 days, IL-1 β gene induction occurred but returned to basal level by the time of analysis. These informations have been added to the corresponding Results section to account for the apparent discrepancy between IL-1 β data in Fig. 5C and 5F.

- Figure 6 - a number of antibodies are used - CD49f, CD138, CD19, etc. Could the authors please indicate what cell types these antibodies are staining, at least in the figure legend? The targets of antibodies used to stain murine cells and skin sections are now specified in the legends of Fig. 6 and Fig. S5 and corresponding result section.

References

- Byrd AS, Carmona-Rivera C, O'Neil LJ, Carlucci PM, Cisar C, Rosenberg AZ, Kerns ML, Caffrey JA, Milner SM, Sacks JM, *et al* (2019) Neutrophil extracellular traps, B cells, and type I interferons contribute to immune dysregulation in hidradenitis suppurativa. *Sci Transl Med*
- Curran AM, Girgis AA, Jang Y, Crawford JD, Thomas MA, Kawalerski R, Collier J, Bingham CO, Na CH & Darrah E (2023) Citrullination modulates antigen processing and presentation by revealing cryptic epitopes in rheumatoid arthritis. *Nat Commun*
- Gudjonsson JE, Tsoi LC, Ma F, Billi AC, van Straalen KR, Vossen ARJV, van der Zee HH, Harms PW, Wasikowski R, Yee CM, *et al* (2020) Contribution of plasma cells and B cells to hidradenitis suppurativa pathogenesis. *JCI Insight*
- Guet-Revillet H, Coignard-Biehler H, Jais JP, Quesne G, Frapy E, Poirée S, Le Guern AS, Le Flèche-Matéos A, Hovnanian A, Consigny PH, *et al* (2014) Bacterial pathogens associated with hidradenitis suppurativa, France. *Emerg Infect Dis*
- Guet-Revillet H, Jais JP, Ungeheuer MN, Coignard-Biehler H, Duchatelet S, Delage M, Lam T, Hovnanian A, Lortholary O, Nassif X, *et al* (2017) The Microbiological Landscape of Anaerobic Infections in Hidradenitis Suppurativa: A Prospective Metagenomic Study. *Clin Infect Dis*
- Hotz C, Boniotto M, Guguin A, Surenaud M, Jean-Louis F, Tisserand P, Ortonne N, Hersant B, Bosc R, Poli F, *et al* (2016) Intrinsic Defect in Keratinocyte Function Leads to Inflammation in Hidradenitis Suppurativa. *J Invest Dermatol*
- de Jongh CA, de Vries TJ, Bikker FJ, Gibbs S & Krom BP (2023) Mechanisms of Porphyromonas gingivalis to translocate over the oral mucosa and other tissue barriers. *J Oral Microbiol* doi:10.1080/20002297.2023.2205291 [PREPRINT]
- K L, A D & L S (2024) Secreted Proteases from Vaginal Prevotella Species Target and Mimic Human MMPs and Modulate Endocervical Barrier Function. *Am J Obstet Gynecol*
- Matsushima H & Takashima A (2009) Real-time in vivo imaging of IL-1 beta producing cells in inflamed murine skin. *J Invest Dermatol*
- Tohgasaki T, Ozawa N, Yoshino T, Ishiwatari S, Matsukuma S, Yanagi S & Fukuda H (2018) Enolase-1 expression in the stratum corneum is elevated with parakeratosis of atopic

dermatitis and disrupts the cellular tight junction barrier in keratinocytes. *Int J Cosmet Sci*

12th Nov 2025

Dear Prof. Demangel,

Thank you for submitting your revised manuscript to EMBO Molecular Medicine, and please accept my apologies for the delay in getting back to you as one referee needed more time to complete their evaluation. As you will see below, while referees #1 and #2 are overall satisfied with the revisions, referee #3 still raises a few concerns that should be addressed in an additional round of revisions. Please also address the remaining minor comments from referee #1.

As EMBO Press usually only allows one round of revisions, please be aware that this will be your last opportunity to address these issues.

Moreover, please address the following editorial requests:

1. Please indicate the modifications in your manuscript in track changes mode.
2. Please upload your EV figures as individual files. Please change the heading "Supplementary Information" to "Expanded View Figure Legends" in the manuscript file. Your supplementary figures are identical to the EV figures, with the exception of figure EV2. Please check and clarify. If you do have an Appendix, please add a table of contents with page numbers. Please also add figure legends and place them underneath the corresponding figures. Please correct the nomenclature to 'Appendix Figure S1', etc, throughout, and add their callouts in the manuscript text.
3. Please note that all corresponding authors are required to supply an ORCID ID for their name upon submission of a revised manuscript. An ORCID identifier is currently missing for O. Join-Lambert.
4. Data Availability: Primary datasets produced in this study need to be deposited in an appropriate public database. In case you have no data that requires deposition in a public database, please state: "This study includes no data deposited in external repositories". Please remove "Bacterial isolates are available from OJL with a completed material transfer agreement and information about the presented data from corresponding authors without restriction. No samples nor animals were excluded from the analysis."
5. Author contributions: CRediT has replaced the traditional author contributions section. Please remove the Authors Contributions from the manuscript and use the free text boxes beneath each contributing author's name in our system to add specific details on the author's contribution. More information is available in our guide to authors.
6. "Conflict of interest" should be renamed "Disclosure and competing interests statement".
7. Thank you for providing a nice visual abstract. Please upload it as a tiff/jpeg/PNG file 550 px wide x 300-600 px high. A cropped portion of this image will serve as thumbnail for the table of content on our webpage.
8. Please address the queries from our data editors in the figure legends:
 - Please note that the exact p values are not provided in the legends of figures 1A, D; A, B; 5A, C, F; 6B, EV1 A
 - Please indicate the statistical test used for data analysis in the legend of figure 1A
 - Please note that information related to n is missing in the legends of figures 1A, D; 2B, 3A, B; EV1 A, EV2 B

As part of the EMBO Publications transparent editorial process initiative (see our Editorial at <http://embomolmed.embopress.org/content/2/9/329>), EMBO Molecular Medicine will publish online a Review Process File (RPF) to accompany accepted manuscripts.

This file will be published in conjunction with your paper and will include the anonymous referee reports, your point-by-point response and all pertinent correspondence relating to the manuscript. Let us know whether you agree with the publication of the RPF and as here, if you want to remove or not any figures from it prior to publication.

I look forward to receiving your revised manuscript.

With kind regards,

Lise Roth

***** Reviewer's comments *****

Referee #1 (Remarks for Author):

The authors have replied to the comments raised. However, I would still suggest to present the data in Figure 2D, as AUC graphs (similar to what is now presented in 2C for LZ, but comparing also with and without depletion). Raw ELISA curves can be then shown in the supplementary figures. Otherwise, it is still unclear what is a negative loss of IgG post depletion. Also currently, S2C figure shows data on SIgA and not on LZ.

Referee #2 (Remarks for Author):

is suitable for publication

Referee #3 (Comments on Novelty/Model System for Author):

Some technical issues remain (outlined below)

Referee #3 (Remarks for Author):

While most of my critiques were adequately addressed, there are some remaining issues that should be resolved:

1. Regarding the patient cohorts, and corresponding metagenomic, serum, and skin data -- Even if metagenomic sequencing was performed on only HS patients, the full relative abundance profiles could still be shown across the 3 stages. Given the low relative abundances of the bacteria that the authors chose to further investigate, and the fact that these isolated bacteria exist as part of a broader microbial community, this information is important context for the reader.
2. It is problematic to compare an all female control group to an all male experimental group (Table 3).
3. The authors conclude that they do not detect *P. uenonis* penetrating ear skin, because it must have happened at a time point that was not captured. To me, this is not a satisfying answer. Was the bacteria sitting on top of the skin, if not penetrating? Is it possible that the bacteria is colonizing the gut, resulting in the immune responses observed at day 60 post TA? The methods state 1 mL of culture ($\sim 10^7$ CFU, but not exact since only OD's were calculated) was applied to the ear, but 1 mL of liquid on a mouse is a lot, even if the entire mouse (and not just the ear) was colonized. The bacteria could have been ingested and reside in the gut. Overall, there needs to be more clarity about what happens to the bacteria in these mouse experiments.

Referee #1 (Remarks for Author):

The authors have replied to the comments raised. However, I would still suggest to present the data in Figure 2D, as AUC graphs (similar to what is now presented in 2C for LZ, but comparing also with and without depletion). Raw ELISA curves can be then shown in the supplementary figures. Otherwise, it is still unclear what is a negative loss of IgG post depletion. Also currently, S2C figure shows data on SIgA and not on LZ.

Figure 2D now includes a panel showing the reactivity of HC and HS3 sera pools against LZ following depletion on *P. uenonis* or *P. bivia*, expressed as AUC determined by ELISA with serially diluted sera. The corresponding ELISA titration curves are provided in EV2C.

Referee #3 (Remarks for Author):

1. Regarding the patient cohorts, and corresponding metagenomic, serum, and skin data, even if metagenomic sequencing was performed on only HS patients, the full relative abundance profiles could still be shown across the 3 stages. Given the low relative abundances of the bacteria that the authors chose to further investigate, and the fact that these isolated bacteria exist as part of a broader microbial community, this information is important context for the reader.

We have added an additional panel to Fig. 1A to provide this information for readers.

2. It is problematic to compare an all female control group to an all male experimental group (Table 3).

We agree that comparing an all-female control group with an all-male experimental group is a limitation. Unfortunately, additional HS3 samples were not available. To address this concern, we included skin samples from two additional male healthy controls and performed ISH for *Pu*. No *Pu* signal was detected in either of these male controls, whereas one of the previously tested HS3 samples showed positive staining (representative image shown below). Table 3 and the corresponding text have been updated accordingly.

3. The authors conclude that they do not detect *P. uenonis* penetrating ear skin, because it must have happened at a time point that was not captured. To me, this is not a satisfying answer. Was the bacteria sitting on top of the skin, if not penetrating? Is it possible that the bacteria is colonizing the gut, resulting in the immune responses observed at day 60 post TA? The methods state 1 mL of culture ($\sim 10^7$ CFU, but not exact since only OD's were calculated) was applied to the ear, but 1 mL of liquid on a mouse is a lot, even if the entire mouse (and not just the ear) was colonized. The bacteria could have been ingested and reside in the gut. Overall, there needs to be more clarity about what happens to the bacteria in these mouse experiments.

We thank the reviewer for raising these important questions and provide below a more detailed explanation of the data supporting our conclusion.

We attempted to detect skin-colonizing bacteria 7, 14, and 60 days after association using both skin swabs (sampling the surface) and skin tissue digests (capturing bacteria that had colonized skin appendages). Samples were plated on agar and incubated under anaerobic conditions. The colony morphology did not resemble *P. bivia* or *P. uenonis*, and 16S rDNA Sanger sequencing confirmed their absence. Since neither swabs nor tissue digests indicated the presence of the topically associated bacteria, we concluded that *P. uenonis* and *P. bivia* do not survive in intact mouse skin or persist at levels too low for detection by culture-based methods. This is consistent with the biology of both species, which are anaerobes and do not naturally persist in the intact skin microenvironment.

The reviewer also raised an important point regarding the bacterial culture used for topical association. To clarify, colony-forming units (CFU) were determined for the optical density

(OD) used in the topical association assays. Once the most favorable conditions were identified, 1 mL of culture was applied to healthy, intact mouse ear skin. We have previously shown that this type of skin tissue association does not cause infection nor compromise the skin barrier. Naturally skin-homing microbes, such as *Staphylococcus epidermidis* (Gribonika et al., *Nature*, 2025; Naik, Bouladoux et al., *Nature*, 2015) and *Staphylococcus aureus* (Enamorado et al., *Cell*, 2023), colonize the host at very low abundance, highlighting the commensal nature of this method. Mouse ear skin already hosts an endogenous microbiome, which actively prevents colonization by “outsider” microbes, including potential pathogens. Consequently, even with naturally skin-homing bacteria, recovered CFUs are consistently very low.

Finally, the reviewer suggested that some bacteria might be ingested and colonize the gut. While we did not assess gut colonization, we agree this is possible. As noted above, only a very limited number of bacteria used for skin association establish residency in the skin. Most of the applied culture is likely cleared via grooming or shedding and may be distributed in the cage environment or ingested. Nevertheless, the strong presence of *P. uenonis*-specific plasma cells in the skin demonstrates that an immune response is initiated in response to the skin colonization event, indicating significant exposure of bacterial antigens within the cutaneous microenvironment. We hope that these explanations address the reviewer’s questions. We have also updated the manuscript accordingly in the Results and M&M sections to reflect these clarifications.

13th Feb 2026

Dear Prof. Demangel,

Thank you for submitting your revised study. We have now received the report from referee #3, who is satisfied with the revisions. I will therefore be able to accept your manuscript once the following minor editorial matters are addressed:

1/ Manuscript text:

- Please accept the previous changes and only indicate in track changes mode any new modification.
- "Material and Methods" should be renamed "Methods":
 - o Human samples: please include a statement confirming that the experiments conformed to the principles set out in the WMA Declaration of Helsinki and the Department of Health and Human Services Belmont Report.
 - o Cells: please indicate whether the cells were tested for mycoplasma contamination.
 - o Animals: if applicable, please provide the reference number for ethics approval.
- Data availability: Please remove the current text. Metagenomics data should be deposited in an appropriate repository, for instance NCBI.

2/ Figures:

- Please remove the legends from the EV figures files.
- Figure 4E: there is a white line displayed in the middle of the right bottom picture, please check and correct if needed.
- Please address the queries from our data editors in the figure legends:
 - n should be defined in the legends of figures 1A, D; 2B, 3A, B; EV1 A, EV2 B.

3/ Checklist: please fill in the section on Cell materials/ mycoplasma contamination. Please also complete the section on data deposition.

4/ As part of the EMBO Publications transparent editorial process initiative (see our Editorial at <http://embomolmed.embopress.org/content/2/9/329>), EMBO Molecular Medicine will publish online a Review Process File (RPF) to accompany accepted manuscripts.

This file will be published in conjunction with your paper and will include the anonymous referee reports, your point-by-point response and all pertinent correspondence relating to the manuscript. Let us know whether you agree with the publication of the RPF and as here, if you want to remove or not any figures from it prior to publication.

I look forward to receiving your revised manuscript.

Yours sincerely,

Lise Roth

***** Reviewer's comments *****

Referee #3 (Comments on Novelty/Model System for Author):

No concerns

Referee #3 (Remarks for Author):

I am satisfied with the revisions made

The authors addressed the remaining editorial issues.

3rd Mar 2026

Dear Prof. Demangel,

Thank you for submitting your revised files. I am pleased to inform you that your manuscript is accepted for publication and is now being sent to our publisher to be included in the next available issue of EMBO Molecular Medicine.

Thank you for depositing your metagenomic data in NCBI. Please make them publicly available and provide a URL at proof stage.

You may qualify for financial assistance for your publication charges - either via a Springer Nature fully open access agreement or an EMBO initiative. Check your eligibility: <https://link.springer.com/journal/44321/how-to-publish-with-us>

Yours sincerely,

>>> Please note that it is EMBO Molecular Medicine policy for the transcript of the editorial process (containing referee reports and your response letter) to be published as an online supplement to each paper. If you do NOT want this, you will need to inform the Editorial Office via email immediately. More information is available here: <https://link.springer.com/partners/embo-press/editorial-policies#Peer%20review>